# Bacterial chromatin remodeling associated with transcription-induced domains at pathogenicity Islands

Mounia Kortebi[1], Mickaël Bourge [1], Romain Le Bars [1], Erwin Van Dijk[1], Charles J. Dorman [2], Stéphanie Bury-Moné [1], Frédéric Boccard [1] & Virginia S. Lioy [1]✉

The nucleoid-associated protein H-NS is a bacterial xenogeneic silencer responsible for preventing costly expression of genes acquired through horizontal gene transfer. H-NS silences several *Salmonella* Pathogenicity Islands (SPIs) essential for host infection. The stochastic expression of SPI-1 is required for invasion of host epithelial cells but complicates investigation of factors involved in SPI-1 chromatin structure and regulation. We performed functional genomics on sorted *Salmonella* populations expressing SPI-1 or not, to characterize how SPI-1 activation affects chromatin composition, DNA conformation, gene expression and SPI-1 subcellular localization. We show that silent SPIs are associated with spurious antisense transcriptional activity originating from H-NS-free regions. Upon SPI-1 activation, remodeling of H-NS occupancy defines a new chromatin landscape, which together with the master SPI-1 regulator HilD, facilitates transcription of SPI-1 genes. SPI-1 activation promotes formation of Transcription Induced Domains accompanied by repositioning SPI-1 close to the nucleoid periphery. We present a model for tightly regulated chromatin remodeling that minimizes the cost of pathogenicity island activation.

Bacterial chromosomes are organized in a dynamic and multiscale manner by Nucleoid Associated Proteins (NAPs), condensin complexes and various biological processes[1–3]. NAPs are master operators working at different scales of nucleoid organization and function; they not only control DNA conformation and DNA condensation through binding, bending, wrapping or bridging DNA molecules, but they also control gene expression, acting like general transcriptional factors through binding to promoters or by silencing gene expression[4]. The NAP H-NS has received considerable attention as it prevents the expression of horizontally acquired AT-rich genes by polymerizing along extended DNA regions. H-NS, acting at different steps of the transcription process, can repress hundreds of gene targets[5–8]. In vivo, ChIP-on-ChIP experiments performed in *Salmonella enterica* serovar Typhimurium (hereafter *Salmonella*) revealed that H-NS interacts with extensive

stretches of DNA in regions rich in exogenous sequences that code for virulence genes, known as *Salmonella* Pathogenicity Islands (SPIs)[8]. By binding to SPIs, H-NS is able to silence these horizontally acquired virulence genes, so that they can be incorporated into the genome in an 'inactive state' that imposes a lower fitness burden on the host[7–9]. Reversing the transcriptional silencing activity of H-NS can be achieved by physico-chemical factors (i.e., osmolarity, pH or temperature)[10], antagonistic proteins that interfere with the H-NS/DNA filaments, or through alteration of DNA topology (bending or supercoiling), leading to specific derepression[10–14].

The impact of H-NS binding on chromatin conformation was studied in vivo in *Escherichia coli* using Hi-C[15]. Hi-C is a genome-wide technique that reveals the network of DNA contacts promoted by DNA binding proteins, allowing the 3D organization of bacterial

[1]Université Paris-Saclay, CEA, CNRS, Institute for Integrative Biology of the Cell (I2BC), Gif-sur-Yvette, France. [2]Department of Microbiology, Moyne Institute of Preventive Medicine, Trinity College Dublin, Dublin 2, Ireland. ✉e-mail: virginia.lioy@i2bc.paris-saclay.fr

chromosomes to be studied with high resolution[16]. This technique revealed that local binding of H-NS prevents a large fraction of H-NS targets from interacting with their neighboring loci, without a major impact on global chromosome folding[15]. Nevertheless, in some particular targets, H-NS can form in vivo bridged DNA structures that silence gene expression[17]. Interestingly, the bridging activity of H-NS was described as responsible for transposon capture in *Acinetobacter baumannii*[18] and also in increasing the cohesion between sister chromatids at the *Vibrio* Pathogenicity Island 1 and O-antigen regions[19], unveiling that H-NS-promoted bridging of DNA molecules can also be inter-molecular.

*Salmonella* SPI-1 is a 43 kb locus encoding the Type III Secretion System 1 (T3SS1) and effector proteins essential for *Salmonella* invasion of host cells and modulation of the host immune response[20]. The genes in SPI-1 exhibit stochastic patterns of expression essential for optimal epithelial cell invasion[21,22]. The SPI-1$^{OFF}$ to SPI-1$^{ON}$ switch is controlled by the SPI-1 encoded master regulator HilD[20]. HilD expression is tightly regulated at the transcriptional, post-transcriptional, translational and post-translational levels[23–30]. While H-NS represses *hilD* transcription[29], its translation is secured by different factors that bind to the *hilD* mRNA, including the small regulatory RNA (srRNA) Spot-42[24]. HilD is negatively regulated at the post-translational level by direct interaction with HilE[27,30]. A recent model proposed that spurious transcription from *hilD* secondary promoters allows the stochastic expression of the *hilD* gene, which counter-silences H-NS at the *hilD* main promoter, initiating the SPI-1 activation cascade[14,31].

Here, we characterize SPI chromatin organization on sorted populations of *Salmonella* that do or do not express SPI-1. Silenced SPI chromatin does not form domains and is correlated with spurious antisense transcriptional activity arising from H-NS-free regions. Upon expression, SPI-1 chromatin is remodeled into Transcription-Induced Domains[32] (TIDs) in a HilD-dependent manner. In these TIDs, the H-NS occupancy is largely reorganized, and coding regions free of H-NS are associated with increased transcriptional activity and a characteristic SPI-1 expression signature. Three-Dimensional Structured Illumination Microscopy[33] (3D-SIM) revealed that SPI-1 TIDs are relocated to the nucleoid periphery. We propose that the presence of regions with unchanged H-NS occupancy at the borders of SPI-1 might act as areas of H-NS confinement, essential for rapid SPI-1 repression upon subcellular relocation. Altogether, we show that the tight regulation and functional activation of SPI-1 involve chromatin remodeling of the entire region and subcellular relocation closer to the periphery of the nucleoid.

## Results

### Transcriptomic analysis in sorted populations reveals a new SPI-1 signature

*Salmonella* SPI-1 is bistably expressed in early stationary phase (ESP)[21,34] in about 10% of cells (Supplementary Fig. 1a, Table 1). To determine the large-scale organization of the *Salmonella* chromosome and investigate the chromatin organization of horizontally acquired pathogenicity islands in silenced *vs* active conditions, we developed a method to perform functional genomics in bacterial populations in which SPI-1 is either OFF or ON (Fig. 1a, Supplementary Fig. 1a), using a well characterized reporter system[21] in which the *gfp* gene is under the control of the *prgH* promoter (P$_{prgH}$). This promoter controls the expression of *prgH*, a gene encoding a component of the type three secretion apparatus of *Salmonella*. The reporter cassette cloned in single copy at the *putPA* locus[21] was used to sort SPI-1$^{ON}$ (GFP$^+$) cells from SPI-1$^{OFF}$ (GFP$^-$) cells. After cell sorting, GFP$^+$ cells represented more than 80% of cells in the GFP$^+$ population and ~0.03% of cells in the GFP$^-$ population (Supplementary Fig. 1a).

To validate our approach, we analyzed the transcriptomic profiles of the sorted populations. Out of 84 differentially expressed genes (Methods, Supplementary Data 1), 71 were highly expressed and 13 were repressed in the GFP$^+$ population (Fig. 1b, Supplementary Data 1). Our transcriptomic data were compared with those obtained previously in heterogeneous populations expressing or not SPI-1 (ESP *vs* early exponential phase (EEP) condition[34]) and in GFP$^+$ *vs* GFP$^-$ sorted populations. This comparison revealed that 60 of the genes upregulated in the GFP$^+$ population were also upregulated in ESP (Supplementary Fig. 1b). Furthermore, 39 of the upregulated genes in GFP$^+$ cells correspond to SPI-1 genes (30 genes) or other SPI genes (SPI-5: 2 genes; SPI-11: 2 genes, SPI-2: 1 gene; SPI-3: 1 gene; SPI-4: 3 genes), most of them known to be co-regulated with SPI-1[34] (Fig. 1b, and Supplementary Table 1, Supplementary Table 2 and Supplementary Data 1). In addition, we found 11 genes co-expressed with SPI-1 that had not been identified previously (Table 2). Among them, the *spf* gene, previously identified as downregulated when SPI-1 is expressed[34] (Supplementary Fig. 1b, Supplementary Data 1), codes for an srRNA known to bind to the 3′ UTR of *hilD* mRNA, preventing degradation of the *hilD* transcript during SPI-1 expression[24]. Upregulation of *spf* might thus be important for *hilD* mRNA stabilization in *Salmonella* populations expressing SPI-1. In addition, we identified 8 new downregulated genes in GFP$^+$ *Salmonella* cells (Table 3).

### Active SPI-1 chromatin locally remodeled into HilD-dependent TIDs

To determine the global folding of the *Salmonella* chromosome in sorted populations we established Hi-C contact maps at 1 kb bin resolution (Fig. 1c). The Hi-C map shows that the overall 3D organization of the chromosome in the GFP$^-$ subpopulation is similar to that of *E. coli*[15], with the presence of a constrained ~600-kb long region *(ter)* centered on the terminus of replication. Outside this region, DNA contacts can extend up to ~1 Mb (Fig. 1c, Figure Supplementary 1c). The folding of the *Salmonella* chromosome in SPI-1 silent populations is conserved in exponentially growing cells and in other infection relevant conditions in which SPI-1 is bistably expressed (Supplementary Fig. 1d). Remarkably, Hi-C contact maps obtained from GFP$^+$ sorted populations revealed no major changes in the global folding of the genome, only local changes in the vicinity of the active SPI-1 locus, delineated by a strong boundary (Fig. 1c and Supplementary Fig. 2a).

We inspected in more detail the Hi-C contact map at SPI loci in the sorted populations (Fig. 2a and Supplementary Fig. 2b-l). Magnification of contact maps centered at SPIs revealed that three of the SPIs in which genes were upregulated (SPI-5, SPI-1 and SPI-4) presented a local increase of contacts within their loci (Supplementary Fig. 2e, Fig. 2a and Supplementary Fig. 2l). Qualitative comparison of the ratio of normalized contact matrices of sorted populations (GFP$^+$/GFP$^-$) confirmed increased frequency of contacts within SPI-5, SPI-1 and SPI-4 loci in the GFP$^+$ population (Supplementary Fig. 2e, Fig. 2e and Supplementary Fig. 2l). Interestingly, the folding of active SPI-1 and SPI-4 is associated with domains, while active SPI-5 forms a loop with itself and its downstream genetic region (Supplementary Fig. 2e). We observed no changes in SPI-11, SPI-2 or SPI-3 chromatin, although some genes were upregulated (Fig. 1b, Supplementary Table 2, Supplementary Data 1, and Supplementary Fig. 2f-2g and 2k). Finally, we did not

## Table 1 | Percentage of GFP$^+$ cells in the different strains and conditions studied, as determined by FACS

| Strain | % GFP+ cells[*] | |
|---|---|---|
| | EEP (OD 0.1) | ESP (OD 2) |
| P$_{prgH}$-*gfp* | 0.7 ± 0.5 | 10.7 ± 1.1 |
| *hilD* P$_{prgH}$-*gfp* | 0.10 ± 0.1 | 0.3 ± 0.2 |
| *hilE* P$_{prgH}$-*gfp* | 1.6 ± 1.1 | 29 ± 3.9 |

* Mean values +/- standard deviation of three independent experiments.

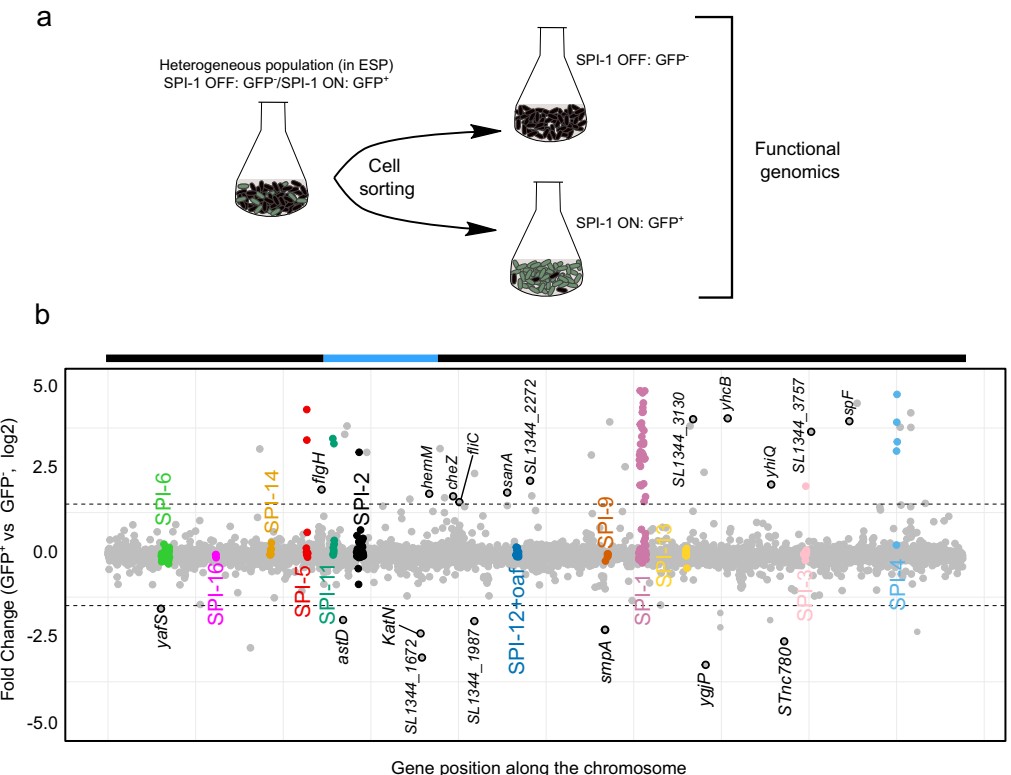

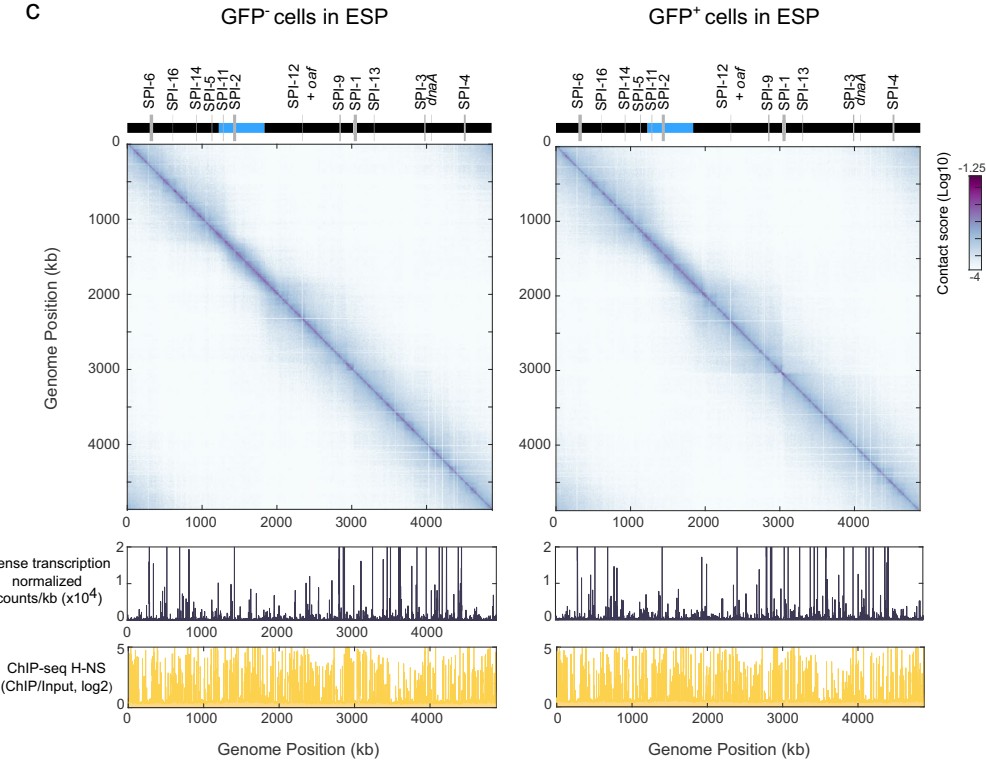

**Fig. 1 | Chromosome organization in GFP⁻ and GFP⁺ *Salmonella* cells.**
**a** Experimental setup to perform functional genomics in sorted populations grown in early stationary phase (ESP) based on the expression of $P_{prgH}$-*gfp*. **b** Fold change expression along the genome of GFP⁺ vs GFP⁻ populations in ESP. Dashed lines show genes with a fold change in expression of three times. SPI genes are highlighted in color and SPI number is indicated in the plot. New regulated genes identified in this study are highlighted in gray and black. The rest of the genes are shown in gray. The upper bar represents the *ter* (blue) and non-*ter* regions (in black). **c** 3D *Salmonella* chromosome folding (top panel), gene expression profile (middle panel) and H-NS protein occupancy in GFP⁻ (left) or GFP⁺ (right) cells in ESP.

**Table 2 | New upregulated genes identified as co-expressed with SPI-1**

| Gene | Encoded product | KEGG Orthology (KO) | KO sub category |
|---|---|---|---|
| *flgH* | Flagellar L-ring protein FlgH | Cellular processes | Cell motility |
| *hemM* | Outer membrane lipoprotein LolB | Not Included in Pathway or Brite | Unclassified: signaling and cellular processes |
| *cheZ* | Chemotaxis protein CheZ | Cellular processes | Cell motility |
| *fliC* | Flagellin | Environmental information processing | Signal transduction |
| *sanA* | Vancomycin resistance protein | Not included in pathway or brite | Poorly characterized |
| *SL1344_2272* | Undecaprenyl phosphate-alpha-L-ara4N flippase sub-unit ArnF | Human diseases | Drug resistance: antimicrobial |
| *SL1344_3130* | Hypothetical protein | Not assigned | Not assigned |
| *yhcB* | Conserved hypothetical protein | Not included in pathway or brite | Poorly characterized |
| *yhiQ* | 16S rRNA (guanine1516-N2)-methyltransferase | Brite hierarchies | Protein families: genetic information processing |
| *SL1344_3757* | Hypothetical protein | Not assigned | Not assigned |
| *spf* | Small regulatory RNA Spot 42 | Brite hierarchies | Protein families: genetic information processing |

**Table 3 | New downregulated genes identified in GFP+ *Salmonella* cells**

| Gene | Encoded product | KEGG Orthology (KO) | KO sub category |
|---|---|---|---|
| *yafS* | Conserved hypothetical protein (potential protein methyltransferase) | Not assigned | Not assigned |
| *astD* | Succinylglutamic semialdehyde dehydrogenase | Metabolism | Amino acid metabolism |
| *SL1344_1987* | Conserved hypothetical protein | Not assigned | Not assigned |
| *smpA* | Outer membrane protein assembly factor BamE | Brite hierarchies | Protein families: signaling and cellular processes |
| *katN* | Manganese catalase | Not included in pathway or brite | Unclassified: metabolism |
| *STnc780* | Small RNA; experimentally verified | Not assigned | Not assigned |
| *SL1344_1672* | Voltage-gated potassium channel | Brite hierarchies | Protein families: signaling and cellular processes |
| *ygjP* | UTP pyrophosphatase | Metabolism | Nucleotide metabolism |

observe changes in H-NS binding occupancy within any active SPI, except SPI-1 (Fig. 2d and Supplementary Fig. 2e, 2f, 2g, 2k and 2l). We therefore focused on SPI-1 chromatin.

To set up the boundaries of active SPI-1 domains, we computed the Frontier Index (FI)[35]. The FI highlights loci with a significant change in the directional bias of interactions within a 1 kb genomic region (bin). A significant change in the bias of downstream (orange) or upstream (green) interactions constitutes a frontier (cf. Figure 2b). A domain is then defined as a genomic region between two frontiers with a change of directionality (green to orange). In GFP- cells, only the region encompassing genes *sitA* (3029 kb) to s*prB* (3034 kb) forms a small silent domain of 5 kb. In the GFP+ contact map, the FI detected several bins with a significant change in the bias of interactions (Fig. 2b) highlighting the presence of multiple domains. These domains, associated with a significant increase in transcriptional activity (Fig. 2c) and decrease in H-NS occupancy (Fig. 2d), will henceforth be referred to as TIDs[32]. Overall, three TIDs encompassing the *hilC-SL1344_2880* genomic region were unveiled (Fig. 2g). These TIDs identify a region of 33 kb with increased frequency of contacts representing practically the whole SPI-1 locus (Fig. 2b, Fig. 2g). Two of the SPI-1 TIDs extend over the coding regions of the transcriptional activators *hilA* and *invF*[20] respectively (Fig. 2a, b and g). When comparing the binding occupancy of the master activator HilD in a non-sorted bistable SPI-1^ON/OFF populations (ESP condition, Supplementary Fig. 3), we observed that the genetic region encompassing the SPI-1 *hilA* and *invF* genes, correspond to the two regions with higher HilD occupancy.

We performed Hi-C on sorted *hilE* deletion mutant cells harboring the same SPI-1 reporter system[21]. The HilE protein is a negative regulator of HilD[27,30]. In the absence of HilE, the number of GFP+ cells was greater by ~3 fold compared to the *hilE*+ background (Table 1), but this increase did not alter global or local folding of the *Salmonella* chromosome in the GFP- or GFP+ populations (Supplementary Fig. 4a). In the *hilE* deletion mutants as in *hilE*+ cells, only SPI-1, SPI-4 and SPI-5 loci in the GFP+ population presented a local increase in contacts (Supplementary Fig. 4e, 4j and 4m). In the absence of HilD[36], SPI-1 is not active and TIDs are not formed (Table 1, Fig. 2f), confirming that HilD is necessary for SPI-1 chromatin remodeling (i.e., changes in the protein landscape and local 3D organization) and the formation of TIDs.

Taken together, these results show that upon expression, the SPI-1 locus sustains a local change in its 3D folding. This change is associated with a new protein landscape and remodeling of silent SPI-1 chromatin into several TIDs encompassing genes encoding the main regulatory proteins. The SPI-1 TIDs are formed in the absence of HilE and in a HilD-dependent manner.

### Silenced SPI chromatin is associated with spurious antisense transcription

It was proposed that the main function of H-NS is to suppress intragenic transcription events[37,38]. Furthermore, a recent study in *Salmonella* showed that, even under repressed conditions, SPI-1 presents a significant level of spurious transcription originating from antisense promoters. This leads to an open window for displacement of H-NS from regulatory regions, allowing HilD to bind its own promoter and activate SPI-1 transcription[31]. To further explore the transcriptomic data obtained from GFP- population, we decided to investigated the level of spurious transcription—defined as the level transcription in sense or antisense direction that arise in repressed conditions— in SPIs genes (Fig. 3). We observed that in most SPIs, antisense transcription

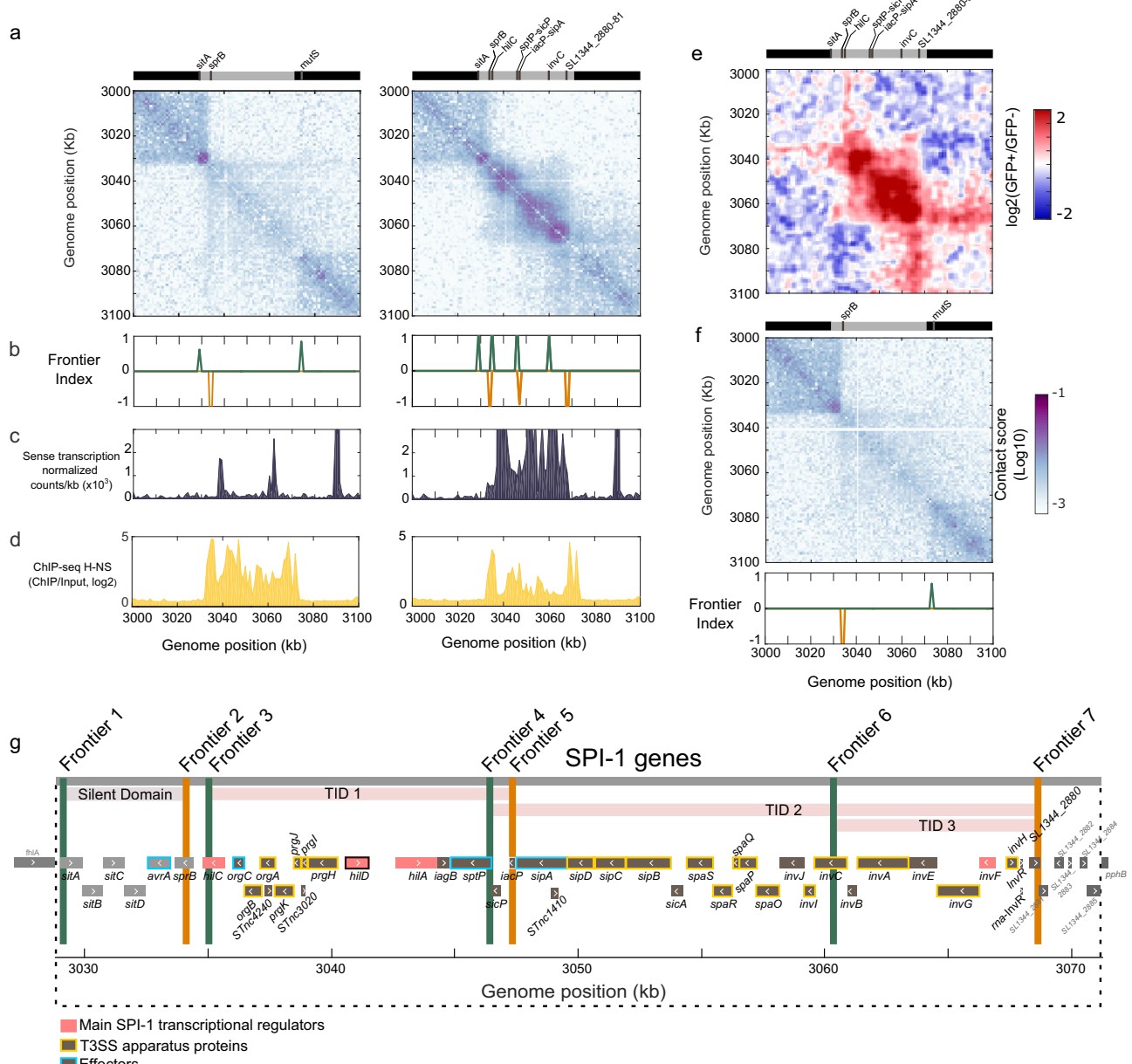

**Fig. 2 | SPI-1 chromatin organization in GFP⁺ or GFP⁻ populations.**
**a** Magnification of Hi-C contact maps centered on SPI-1 in GFP⁻ (left) or GFP⁺ (right) populations in ESP. The upper bars show the beginning and the end of the SPI-1 locus (gray bar) and the genes in which a boundary was identified. **b** The Frontier Index highlights the upwards (green) or downwards (orange) bias of interaction for this genomic region in GFP⁻ (left) or GFP⁺ (right) populations in ESP. Frontier Index Source Data are provided as Source Data file. **c** Sense transcriptional activity (normalized counts per kb) in GFP⁻ (left) or GFP⁺ (right) populations in ESP. **d** H-NS occupancy are shown for GFP⁻ (left) and GFP⁺ (right) populations in ESP. **e** Ratio of normalized contact maps for GFP⁺ compared to GFP⁻ cells in ESP, centered on SPI-1. The upper bar shows the beginning and end the SPI-1 locus (gray bar) and genes in which a boundary was identified in the GFP⁺ condition. **f** Magnification of Hi-C contact map centered on SPI-1 in ESP condition in the absence of HilD. The upper bar shows the beginning and end of the SPI-1 locus (gray bar) and genes in which a frontier was identified. **g** Schematic representation of SPI-1 genes. Genes that encode main SPI-1 transcriptional regulators are colored in pink, with the master SPI-1 regulator *hilD* highlighted in pink and black. Genes encoding for proteins that are part of the T3SS apparatus are highlighted in yellow and genes encoding for effectors in light blue. Green and orange vertical lines represent the position of the frontiers shown in (**b**). The beginning and end of the silent domains and TIDs are indicated with a gray bar and pink bars, respectively.

(as determined by the antisense (AS) index, Fig. 3a, blue boxplots) is significantly higher than or similar to that observed in the *R*est *o*f the *G*enes of the chromosome (RoGs, Fig. 3a, blue boxplots). Interestingly, the overall level of antisense transcription in silent SPIs is significantly higher than in non-SPIs genes (Supplementary Fig. 5a). Concerning the sense transcriptional activity, despite that in SPI-16, SPI-5, SPI-11, SPI-9, SPI-1, SPI-13, SPI-1 and SPI-4, sense transcriptional activity was similar to that in RoGs (Fig. 3b, blue boxplots), the overall sense transcription is

significantly lower in the totality of SPI genes compared to non-SPI genes (Supplementary Fig. 5b). To better understand the origin of the spurious transcription associated with SPIs, we investigated the transcriptional activity depending on whether genes were bound or not by H-NS (Supplementary Fig. 5c and 5d). These results show that antisense transcription is significantly lower in SPI genes that are bound by H-NS, suggesting that this NAP is able to repress the high spurious antisense activity within SPIs. Furthermore, the level of sense transcriptional

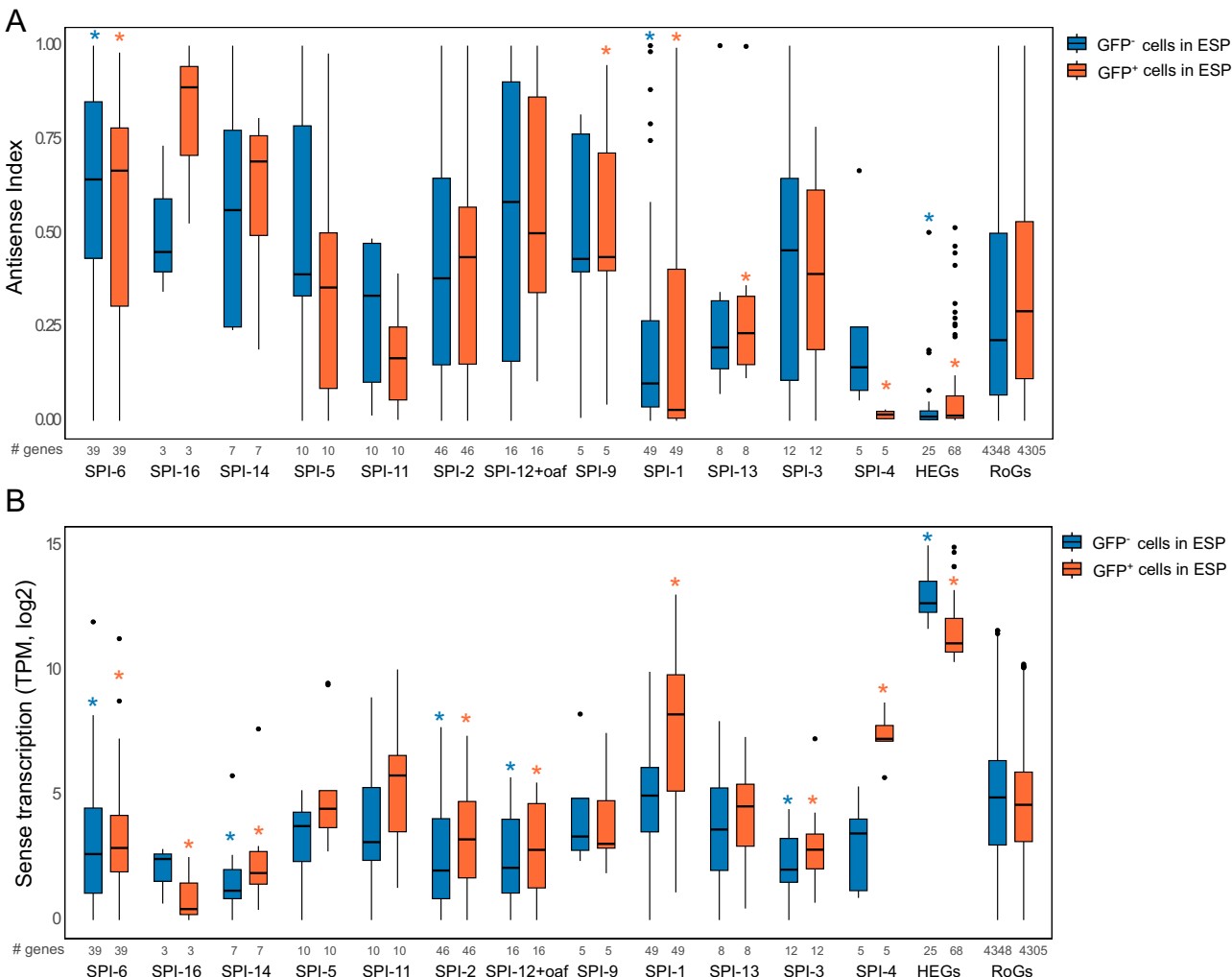

**Fig. 3 | H-NS occupancy and transcriptional activity within SPIs. a** Antisense transcriptional activity within SPIs and highly expressed genes in GFP⁻ (blue) or GFP⁺ (orange) populations in ESP. Features with a significantly different level of transcription than the rest of the genes on the chromosome (RoGs) in the same condition are highlighted with a star in the corresponding colour. **b** Sense transcriptional activity within SPIs and highly expressed genes (HEGs) in GFP⁻ (blue) or GFP⁺ (orange) populations in ESP. Features with a significantly different level of transcription than the rest of the genes on the chromosome (RoGs) in the same condition are highlighted with a star in the corresponding colour. Pairwise comparisons were performed using the two-sided Wilcoxon rank sum test with continuity correction to assess the statistical significance ($p < 0.05$). The p-values are reported in Supplementary Data 3. All boxplots in this figure represent the first quartile, median and third quartile. The upper whisker extends from the hinge to the largest value no further than 1.5* the inter-quartile range (IQR, i.e., distance between the first and third quartiles) from the hinge. The lower whisker extends from the hinge to the smallest value at most 1.5*IQR of the hinge. The number of analysed genes ('#') per feature is indicated in each panel.

activity in the SPI silenced population is significantly lower than in non-SPI genes, independently of being bound by H-NS or not.

We also analyzed transcriptional activity within SPIs in the GFP⁺ population. Analysis of the level of transcription in repressed SPIs again revealed a significantly high spurious transcription in antisense direction (Fig. 3a, orange boxplots and Supplementary 5e). As expected, in active SPIs and in highly expressed genes that are frequently associated with TIDs and not related to SPIs (HEGs), sense transcriptional activity is significantly higher than in RoGs and AS activity decreased (Fig. 3b and Supplementary Fig. 5f).

Altogether, these results highlight that silent SPI genes are associated with a low, but significant, level of spurious antisense transcription compared to non-SPI genes, and that this activity seems to emerge from H-NS free regions.

### H-NS remodeling at SPI-1 during transcriptional activation

We investigated occupancy and the effects of H-NS on chromatin organization in sorted populations by ChIP-seq and Hi-C. In repressed conditions, SPIs are bound by H-NS (Fig. 1c, Fig. 2a and 2d, Supplementary Fig. 2), as previously reported[8]. We correlated Hi-C contact signals with the genome-wide DNA binding profile of H-NS, both performed in the GFP⁻ population at 1 kb (bin). We found that regions with increased DNA-DNA contacts (e. g., TIDs) are poorly bound by H-NS (Spearman correlation = -0.39, $p$ value = 9.5e-184, Fig. 4a). This negative correlation is greater when we consider only the Hi-C contact frequencies within SPIs and the occupancy of H-NS in these horizontally acquired regions (Spearman correlation = -0.65, $p$ value = 2.7e-28). This shows that at repressed SPIs, a high density of H-NS binding is associated with a low frequency of intra SPI DNA contacts, suggesting that H-NS acts as an insulator factor within repressed SPIs.

Consistent with the formation of TIDs associated with active transcription[32,39], H-NS occupancy is significantly reduced in SPI-1 when this pathogenicity island is expressed (Fig. 4b). No significant changes were observed in the H-NS occupancy in other SPIs in which some genes were co-expressed with SPI-1 (i.e., SPI-5, SPI-11, SPI-2, SPI-3 and SPI-4) (Supplementary Fig. 2 and Fig. 4b), The absence of TIDs in

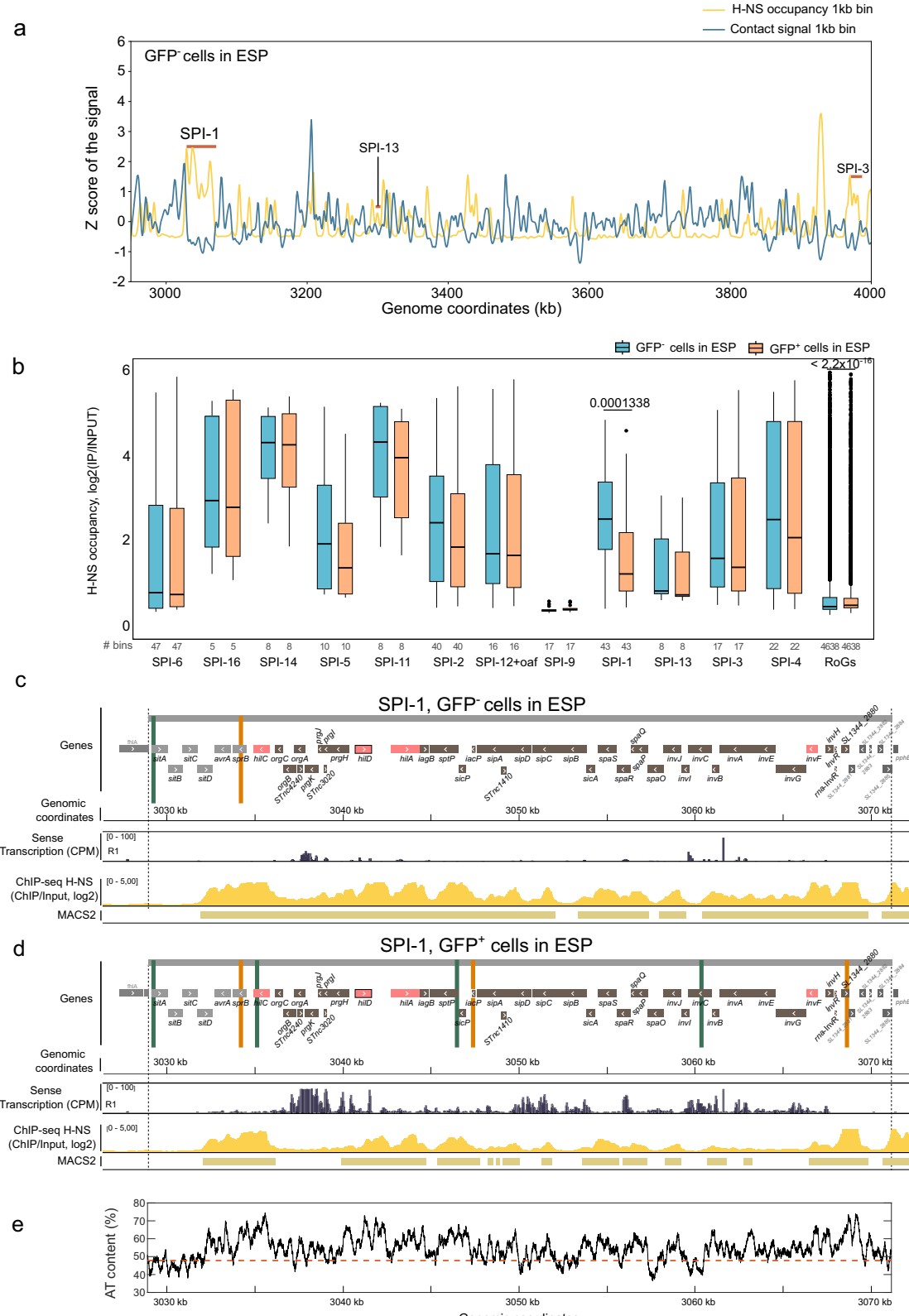

these SPIs suggests that their expression is transient, likely occurring in bursts, leading to the detection of long-lived mRNAs that were expressed concomitant with SPI-1 activation. Additionally, the high occupancy of H-NS within these SPIs confirms the role of this NAP as a chromatin insulator consistent with the absence of TID formation in SPI-5, SPI-11, SPI-2 and SPI-3 or the presence of less well-defined chromatin structure in SPI-4 (Fig. 4b and Supplementary Fig. 2e-g and 2k).

H-NS binding is significantly decreased in active SPI-1, but remains unchanged in other expressed SPIs (Fig. 1c, Fig. 4b and Supplementary Fig. 2b-l). These results suggest that SPI-1 chromatin might be bound by RNA polymerase and/or specific regulatory factors that compete with H-NS binding, leading to a significant reduction in its binding and allowing sustained expression of SPI-1 genes and radical chromatin remodeling.

**Fig. 4 | H-NS occupancy and transcriptional activity in sorted populations of *Salmonella*. a** Magnification of whole-genome correlation between Hi-C contacts at 1 kb bin (blue line) and H-NS occupancy (yellow line) in GFP populations in ESP. SPI-1, SPI-13 and SPI-3 are highlighted with a red line. **b** H-NS occupancy within SPIs in silent (blue) or active (orange) populations in ESP. The significant *p*-values of two-sided Wilcoxon rank-sum tests with continuity correction (SPI-1 OFF vs SPI-1 ON populations) are indicated above each box. The rest of p-values reported in Supplementary Data 4. All boxplots in this figure represent the first quartile, median and third quartile. The upper whisker extends from the hinge to the largest value no further than 1.5* the inter-quartile range (IQR, i.e., distance between the first and third quartiles) from the hinge. The lower whisker extends from the hinge to the smallest value at most 1.5*IQR of the hinge. The number of analyzed bins ('#') per feature is also indicated. **c** In the top panel a schematic representation of SPI-1 genes is shown, indicating the position of frontiers identified in the contact map of GFP populations in ESP. Upwards frontiers are indicated in green and downwards in orange. Main SPI-1 regulators are shown in pink, *hilD* gene is highlighted in black. Below, IGV views of sense transcription (counts per million,'CPM', Replicate 1) and H-NS occupancy on GFP populations in ESP are shown. The numbers between square brackets indicate the scale. Below the ChIP-seq, the MACS2 plot highlights regions bound by H-NS. **d** In the top panel a schematic representation of SPI-1 genes is shown, indicating the position of frontiers identified in the contact map of GFP⁺ populations in ESP. Upwards frontiers are indicated in green and downwards in orange. Main SPI-1 regulators are shown in pink, *hilD* gene is highlighted in black. Below, IGV views of sense transcription (CPM, Replicate 1) and H-NS occupancy on GFP⁺ populations in ESP are shown. The numbers between square brackets indicate the scale. Below the ChIP-seq, the MACS2 plot highlights regions bound by H-NS. **e** Percentage AT content of the DNA within the SPI-1 locus. The orange dotted line indicates the mean AT percentage of the whole chromosome.

As reported above, H-NS ChIP-seq in the GFP⁺ population revealed a significant decrease in the H-NS occupancy exclusively within SPI-1 (Fig. 4b). MACS2[40], a tool developed for identifying DNA regions bound by DNA binding proteins, revealed that H-NS binding in this population shows a variable pattern of occupancy along the SPI-1 locus (Fig. 4c-d). To further understand these differences, we first looked at the A + T content of SPI-1 DNA. Regions rich in AT ( > 55%) presented high H-NS occupancy (Spearman correlation= 0.24, p-value = 5.9e-13, Supplementary Fig. 6a). We observed that regions with an AT percentage lower than 60 % showed a decrease in H-NS occupancy in the SPI-1 locus when this pathogenicity island was expressed (Fig. 4d-e & Supplementary Fig. 6b). Interestingly, in regions with a high AT content (≥ 65%), H-NS showed no or minor changes in binding (Fig. 4d-e, Supplementary Fig. 6b). This is particularly true for regions at the beginning and the end of SPI-1, from genes *avrA* to *orgC* and SL1344_2880 to the end of the island (Fig. 4d-e). Within the *prgH* and *hilD* promoters, with a mean AT percentage of ~65%, H-NS occupancy dropped markedly (Fig. 4d and Fig. 4e and Supplementary Data 2). H-NS occupancy also dropped in the *invF* promoter region, with a median AT content of about 60% and H-NS-free regions were evident when the AT content was between ~55 and 52% (Fig. 4d-e). Interestingly, H-NS-free regions were found all along the central region of SPI-1, the largest ones at the *hilA* and *invF* regulon, specifically within the genes *orgB*, *orgA*, *prgK*, *prgJ*, part of *prgH*, *invG* and part of *invF*. These H-NS-free regions fell within coding sequences in all cases and are associated with intense transcription (Fig. 4d)

### SPI-1 TIDs are relocated towards the nucleoid periphery

It was recently proposed that TIDs are frequently relocated towards the nucleoid periphery, likely to facilitate intense gene expression and translation, since active RNA polymerase and translating ribosomes are mostly excluded from the interior of the nucleoid[32,41–43]. To address the possibility that variable occupancy of H-NS on SPI-1 could be associated with relocation towards the nucleoid periphery to facilitate gene expression, we used Three-Dimensional Structured Illumination Microscopy (3D-SIM) to study the subcellular localization of the SPI-1 locus in *Salmonella* cells. We placed the *parS* site of plasmid pMT1 (*parS^pMT1*)[44] at the end of the SPI-1 locus (see methods), and expressed *parB^pMT1* fused to *yfp* from a multicopy plasmid under the control of the P_lac_ promoter. To identify cells expressing SPI-1, we constructed a transcriptional fusion of the *prgH* gene with the *mCherry* gene at its native position in SPI-1. This system allowed us to visualize simultaneously the SPI-1 locus within cells and the state of SPI-1 (active or repressed) depending on the expression of *mCherry* (Fig. 5a). To determine the relative position of the ParB:YFP/*parS^pMT1* foci with respect to the nucleoid surface, the cells were also stained with Hoechst (Fig. 5a and Fig. 5b). Images were segmented and foci localized in three dimensions (Fig. 5b). Strikingly, after segmentation, many foci were observed outside of the nucleoid (Fig. 5b). A detailed inspection of non-segmented images revealed that, similarly to *E.*

*coli*[42], *Salmonella* nucleoids exhibit a highly dense central DNA mass surrounded by a less dense region, in which ParB:YFP/*parS^pMT1* foci were frequently observed (Fig. 5b, Supplementary Fig. 7). We decided to take the high DNA density region as a reference and expand foci detection in a radius of 0.2 μm (Fig. 5c). Consequently, some foci were detected outside the segmented nucleoid (Fig. 5b-c). After this step, more than 6500 foci were precisely localized in 3D. In order to compare the localizations between the mCherry⁺ (active SPI-1) and mCherry (repressed SPI-1) cells, a threshold was applied based on the mCherry⁺ fluorescent signal (see methods and Supplementary Fig. 8). This approach revealed that ~10% of the ParB:YFP/*parS^pMT1* foci (n = 652) were identified in active SPI-1 cells. In addition, these foci were located more peripherally within the nucleoid when compared to cells in which SPI-1 was repressed (Fig. 5d, Figure Supplementary 9a). Interestingly, while 38.7% of ParB:YFP/*parS^pMT1* foci were detected outside the segmented nucleoids (within the low density nucleoid region) in mCherry cells, this percentage increased to 55.4% in mCherry⁺ cells (Supplementary Fig. 9b). To confirm that changes in subcellular localization were specific to active SPI-1, we analyzed the position of a chromosomal locus unrelated to SPI-1, located 208 kb downstream of this pathogenicity island. Specifically, the *parS^pMT1* was inserted after the stop codon of the gene *yqgF*, which is silent under the ESP condition (Supplementary Data 1 and Supplementary Fig. 9c). The position of *yqgF* was analyzed in a strain carrying the transcriptional fusion of the gene *prgH* with the *mCherry* gene at its native position in the SPI-1 locus, to determine the localization of this locus when SPI-1 was silent (mCherry) or active (mCherry⁺). We did not observe a significant difference in the localization of ParB:YFP/*parS^pMT1* in mCherry⁺ and mCherry cells (Fig. 5d), neither a difference between the percentage of ParB:YFP/*parS^pMT1* foci that were detected outside the segmented nucleoids in mCherry or mCherry⁺ cells (Supplementary Fig. 9c).

All in all, the subcellular localization of SPI-1 in mCherry⁺ and mCherry cells confirms that when SPI-1 is expressed, TID formation is accompanied by a relocation of this locus towards the nucleoid periphery and that it is insulated from the rest of the chromosomal loci.

### Discussion

*Salmonella* and *E. coli* diverged from a common ancestor ~120–160 MYa, when *Salmonella* acquired genes and other mobile elements essential to invade and infect cold-blooded hosts[45,46]. Despite these genetic differences, Hi-C maps revealed that the *S.* Typhimurium chromosome is folded similarly to the *E. coli* chromosome, highlighting the robustness of bacterial genome structure, despite millions of years of evolution.

Our transcriptomic data unambiguously revealed the SPI-1 signature associated with SPI-1 activation. Among the genes upregulated in the GFP⁺ population there is the gene *spf*, which was shown to be downregulated in ESP[24,34](Supplementary Data 1). It is possible that previously, *spf* was detected as downregulated because bistable

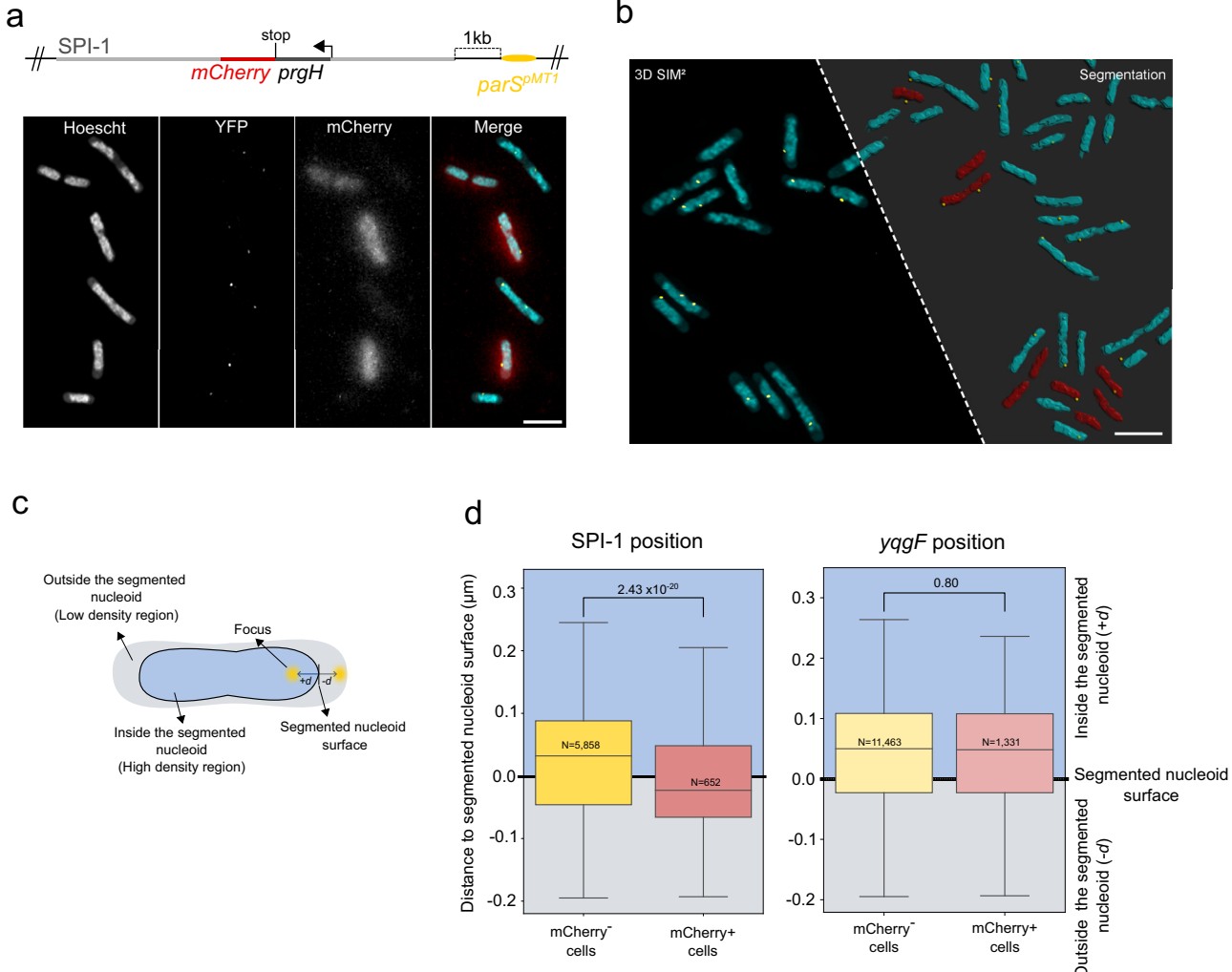

**Fig. 5 | Subcellular localization of SPI-1 in *Salmonella* cells. a** Upper panel: schematic representation of the SPI-1 locus, showing the *prgH(stop)mcherry* transcriptional fusion in its native location and the insertion of *parS^{pMT1}* site, 1 kb upstream of the last SPI-1 gene. Lower panel: Combination of SIM² images of *Salmonella* nucleoids (Hoescht), ParB:YFP/parS^{pMT1} (YFP), wide-field signal of mCherry⁺ and a merge of the three images (maximum intensity projection) in ESP. Scale bar = 2 μm. **b** Representative 3D-SIM field before and after segmentation and identification of active SPI-1 cells in ESP. After segmentation, nucleoids of mCherry⁻ cells are shown in cyan, nucleoids of mCherry⁺ in red and the SPI-1 locus as a green dot. Scale bar = 2 μm. **c** The distance of ParB:YFP/parS^{pMT1} focus is measured from cells by taking the shortest distance to the segmented nucleoid surface, with positive values (+*d*) representing distances inside the segmented nucleoid (high DNA density regions) and negative distances (-*d*) representing distances outside the segmented nucleoid. **D** Box plot representing the distribution of distances for a given

ParB:YFP/parS^{pMT1} focus to the nucleoid surface (black line, '0' value) in mCherry⁻ (yellow) and mCherry⁺ (pink) cells, at the SPI-1 locus (left panel) or the *yqgF* locus (right panel). Positive *y*-axis values represent distances measured for foci that are in the segmented nucleoid (blue), negative values represent distances measured for foci that are outside the segmented nucleoid (gray). All boxplots in this figure represent the first quartile, median and third quartile. The upper whisker extends from the hinge to the largest value no further than 1.5* the inter-quartile range (IQR, i.e., distance between the first and third quartiles) from the hinge. The lower whisker extends from the hinge to the smallest value at most 1.5*IQR of the hinge. The number of analysed foci is indicated in each panel ('N'). Pairwise comparisons were performed using the two-sided Wilcoxon rank-sum test to assess statistical significance and the p-values are shown in each plot. Source data are provided as a Source Data file.

populations were used to analyze SPI-1 co-regulated genes[34]. Thus, *spf* could emerge as a key gene whose transcription could be a signal of pre-adaptation[47] for SPI-1 expression: only those cells in which *spf* repression is bypassed, will express SPI-1. This hypothesis will be the focus of future investigations. Interestingly, we did not observe increased expression of other sRNA known to stabilize *hilD* mRNA[23], suggesting that different sRNAs could target *hilD* mRNA depending on environmental conditions.

Hi-C on *Salmonella* sorted populations expressing SPI-1 or not allowed us to characterize the folding transition of silent to active SPI chromatin. We have shown that the folding of H-NS-bound chromatin does not reveal any particular signature, in support of a model in which H-NS does not influence the large-scale organization of the bacterial

chromosome[15,48]. We did not observe specific bridging between or within repressed SPI loci. It remains possible that H-NS-bridged DNA structures exist but are highly variable from cell to cell and cannot be detected with Hi-C. During the revision of this manuscript, a study using the Micro-C technique, reported that silent *E. coli* chromatin folds into chromosomal hairpins (CHINs) in an H-NS- and StpA-dependent manner[49]. Micro-C is based on chromosome conformation capture (3 C) combined with micrococcal nuclease (MNase) digestion, allowing to stablish contact maps up to 10-bp resolution. It is therefore likely that Micro-C protocol favors the capture of dynamic H-NS/DNA structures that cannot be detected by Hi-C. Our results did show that upon expression, silent SPI-1 is remodeled into several TIDs in a HilD dependent manner. Two of these TIDs include genes of the main

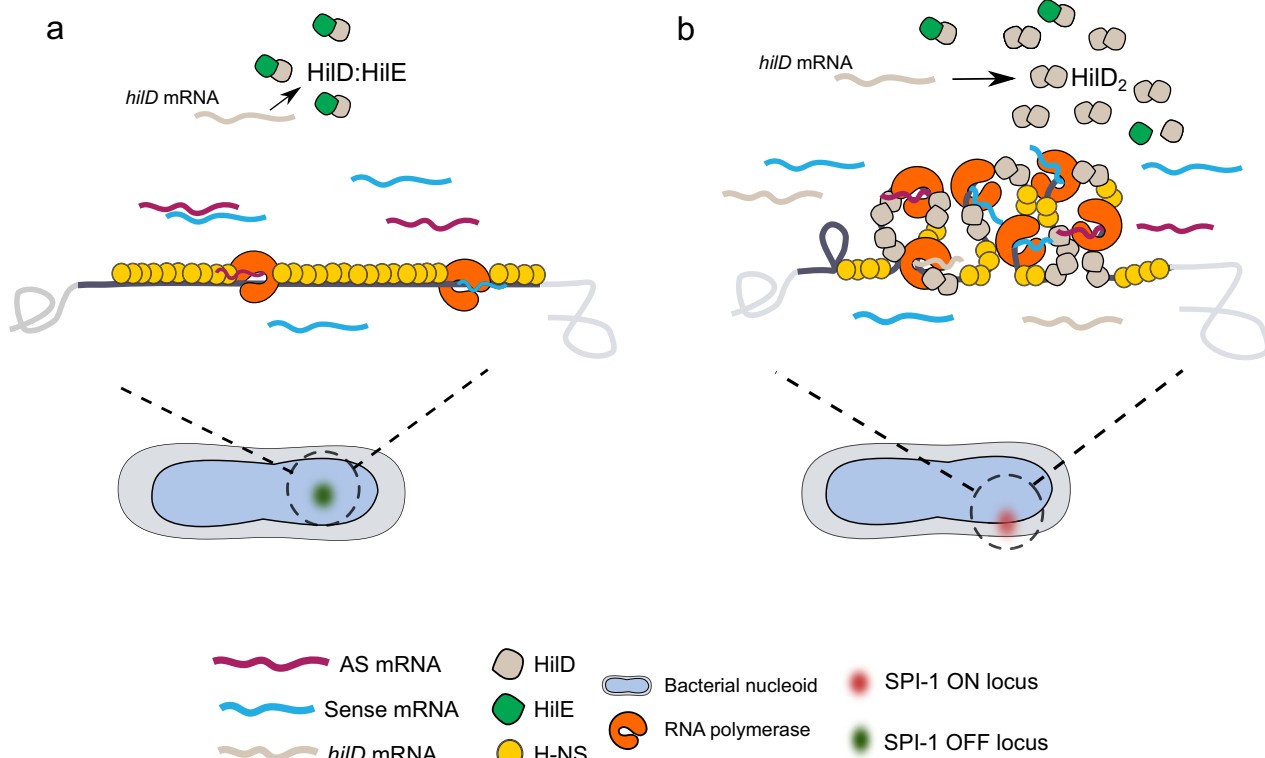

**Fig. 6 | Model of SPI-1 chromatin remodeling upon expression. a** In repressed SPI-1 cells in ESP, SPI-1 chromatin is localized far from the nucleoid periphery, shows a high H-NS occupancy and is permeable to low levels of RNA polymerase molecules that promote the spurious transcription of SPI-1 promoters, such as those of *hilD*. Spurious *hilD* transcription is rapidly degraded. If *hilD* mRNA escapes degradation, the translated HilD protein will form a heterodimer with HilE, impeding the activation of SPI-1. **b** In a fraction of cells in ESP, translation of *hilD* mRNA triggers a feedback loop that activates SPI-1. The binding of HilD to its own promoter leads to a conformational change that promotes the unloading of H-NS, mainly from coding regions, allowing the correct positioning of RNA polymerase and transcript elongation. This is accompanied by the folding of SPI-1 chromatin into several domains and a relocalization close to the nucleoid periphery, where it will facilitate access of RNA polymerase, transcription, translation, assembly, and quick repression of the genes encoding this T3SS.

transcriptional activators of this pathogenicity island (*hilD*, *hilA* and *invF*). In addition, the folding of active SPI-1 chromatin into TIDs seems to be a conserved feature of SPI-1 expression since TIDs are conserved in Hi-C matrices generated from sorted GFP+ populations of a *hilE* deletion mutant. We therefore propose that remodeling of SPI-1 chromatin is essential for faithful gene expression. Our results reveal for the first time that SPI-1 TIDs are associated with H-NS-free regions, the largest corresponding to the *prgH-orgB* and *invF-invE* coding sequences. Finally, we observed that regions with low H-NS occupancy or free of H-NS are associated with increased sense transcriptional activity.

Remarkably, SPI-1 in active chromatin is localized closer to the nucleoid periphery than in silent chromatin. Constraints associated with SPI-1 expression are likely connected with the insulation of SPI-1 TIDs from neighboring loci. The differences in subcellular localization of active and silent SPI-1 lead us to propose that the relocation of SPI-1 towards the nucleoid periphery is the result of high transcription, likely enhanced by the coupling of transcription with translation[43] of the components of the T3SS1 inserted into the cytoplasmic membrane. This interpretation is supported by results in *Vibrio parahaemolyticus* that show a genetic locus encoding a type three secretion system captured at the inner membrane where it is expressed and the T3SS is assembled[50], and by localization of active RNA polymerase at the nucleoid periphery in *E. coli*[42].

Altogether, we propose a model in which a high a density of H-NS co-exists with a low abundance of transcriptionally active RNA polymerase at a silent SPI-1[31], which is mainly located further away from the nucleoid periphery (Fig. 6A). The presence of RNA polymerase will facilitate spurious antisense transcription[51] from H-NS free regions.

Consequently, the low level of antisense transcription might help silence spurious SPI-1 expression. In addition, spurious transcription and high H-NS occupancy are associated with the low frequency of contacts observed within silent SPI-1 loci, forming a linear complex with DNA (Fig. 6a). Spurious *hilD* mRNAs that are produced will lead to production of HilD that will be inactivated by formation of an heterodimer with HilE[27,30]. In contrast, in the GFP+ population, *hilD* mRNA is stabilized, potentially due to its interaction with Spot-42 associated to Hfq[24]. This will lead to sufficient levels of HilD to overcome the negative regulation by HilE, and HilD will be able to bind its own promoter sequence to initiate a feedforward regulatory loop that will lead to SPI-1 activation. The binding of HilD will also promote the counter-silencing of H-NS, particularly from coding sequences with low AT content. Decreased H-NS occupancy from strategic promoter regions will favor transcriptional elongation by RNA polymerase and generation of TIDs within SPI-1 chromatin. Relocation of active SPI-1 at the nucleoid periphery will facilitate the recruitment of new RNA polymerases resulting in high transcription and translation of the entire locus (Fig. 6b). Within this model, the high H-NS occupancy regions observed at the borders of the SPI-1 in active conditions might act as a 'confinement zone' in which H-NS can be stored bound to DNA, avoiding the degradation of free H-NS protein[52] while SPI-1 is expressed, allowing quick repression close to the nucleoid periphery when needed.

In eukaryotes, heterochromatin is highly compacted and inaccessible to transcription factors necessary for gene expression while open chromatin facilitates the access of transcriptional factors and RNA polymerase, promoting gene expression[53]. Here we have shown that bacterial heterochromatin[54], characterized by high H-NS

occupancy, remains permeable to RNA polymerase and does not completely abolish spurious transcriptional activity. In terms of 3D folding, bacterial H-NS-bound heterochromatin is frequently associated with decreased local intra-chromatin interactions (i.e., depletion of specific domains) and increased inter-chromatin interactions[18,19]. The remodeling of active SPI-1 chromatin into TIDs is reminiscent of the eukaryotic process allowing transition from heterochromatin to facultative euchromatin. It also sheds light on domestication of pathogenicity islands[55] by the bacterial host to fine-tune their expression and minimize the fitness cost associated with their activation in a new subcellular location[56].

## Methods

### Bacterial strains, plasmids and growth conditions

All bacterial strains used in this study are derivatives of *Salmonella enterica* serovar Typhimurium strain SL1344[57]. A complete list of the bacterial strains and plasmids used in this study are provided in Supplementary Table 3 and Supplementary Table 4.

Bacteria were cultured in Lennox Broth (LB) medium at 37 °C with shaking at 180 rpm. Overnight cultures were diluted 1:1000 and grown to an OD600 of 0.1 (early exponential phase, EEP) or OD600 of 2 (early stationary phase, ESP), as described previously[34]. When required, antibiotics were added to the liquid medium or agar plates at the following final concentrations: 100 µg/mL ampicillin (Amp), 50 µg/mL kanamycin (Kan), or 30 µg/mL chloramphenicol (Cm).

### Construction of chromosomal mutant strains

The bacteriophage λ Red recombination system was used for the construction of chromosomal deletion mutants and for the fusion of the *3xFLAG* epitope to the C-terminal regions of the H-NS, HilD, and RpoC proteins[58]. Transduction into a clean *S.* Typhimurium SL1344 background was performed using P22 phage[59]. When necessary, resistance cassettes were excised using the pCP20 plasmid[58], which expresses FLP recombinase.

The Δ*hilD* and Δ*hilE* deletion strains were constructed by homologous recombination, replacing the two genes with a chloramphenicol resistance cassette. The pKD3 plasmid was used as a template to amplify the resistance cassette by PCR, using primers containing at their 5′ ends 40-50 bp extensions homologous to the flanking regions of the target gene. The purified PCR product was then introduced into SL1344 strain harboring the pKD46 plasmid to generate the deletion mutants. After PCR verification of the colonies, the mutations were transduced into a clean *S.* Typhimurium SL1344 background.

For all strains carrying the *P*ₚᵣ₉ₕ*gfp* fusion (Supplementary Table 3), the *P*ₚᵣ₉ₕ*gfp* Cmᴿ allele from the SL1344 &*P*ₚᵣ₉ₕ*gfp* Cmᴿ strain[21] was introduced into the respective genetic backgrounds by P22-mediated transduction.

The *prgH(stop)mcherry, parS*ᵖᴹᵀ¹ Cmᴿ strain was generated by inserting a transcriptional fusion of *mCherry* and a kanamycin resistance cassette downstream of the *prgH* stop codon. The template plasmid pMKB1 was derived from pFPV25.1 plasmid[60] (gift from Raphael Valdivia -Addgene plasmid # 20668; (http://n2t.net/addgene:20668); RRID:Addgene_20668), in which the *gfp* gene was replaced with a promoter-less *mCherry* gene from the pFCcGi plasmid[61] (gift from Sophie Helaine & David Holden -Addgene plasmid # 59324; (http://n2t.net/addgene:59324); RRID:Addgene_59324), together with the kanamycin resistance cassette from pKD4[58]. After excision of the kanamycin cassette, the *parS*ᵖᴹᵀ¹ Cmᴿ cassette from plasmid pGBKD3-parSPMT1 was inserted at the end of SPI-1, within the intergenic region between *pphB* and *SL1344_2887*. The strain was subsequently transformed with a plasmid pSPB5, containing the translational fusion *parB*ᵖᴹᵀ¹*-yfp*. Plasmid pSPB5 was constructed using as template plasmid pFH2973[44], in which the genes encoding the fusion *cfp-Δ30parBP1* were deleted by Gibson assembly. Similarly, the *prgH(stop)mcherry,*

*yqgf parS*ᵖᴹᵀ¹ Cmᴿ strain was constructed by inserting the *parS*ᵖᴹᵀ¹Cmᴿ cassette into the intergenic region between *yqgF* and *SL1344_3073*.

### High-throughput chromosome conformation capture (Hi-C)

Bacterial cell crosslinking and Hi-C library preparation were performed as described in ref. 16 with some modifications. Briefly, *Salmonella* cells were grown in LB (see above) in EEP, ESP, oxygen shock or anaerobic growth conditions. A total of ~2 × 10⁸ bacteria were crosslinked with 2.7% formaldehyde (final concentration) in a volume of 10 ml (Sigma-Aldrich, Cat# F8775) and incubated for 30 min at room temperature (RT) with gentle agitation. The excess formaldehyde was quenched by adding 250 mM glycine (Sigma-Aldrich, Cat# G8898) and incubation for 20 min at RT under gentle agitation. The fixed cells were then collected by centrifugation at 4000 × *g* for 10 min and washed twice with 10 ml of 1x PBS (homemade). The cell pellets were stored at −70 °C until further use.

Frozen pellets were thawed on ice, resuspended in 200 µL of 1× Tris-EDTA (TE) buffer (Sigma-Aldrich, Cat# T928) containing a complete protease inhibitor cocktail (Roche, Cat# 11873580001). Then, 1 µl of ready-lyse lysozyme (LGC Biosearch, Cat# LU-R1804M) was added and the mixture was incubated at RT for 20 min. Next, 10 µl of 10% SDS (Sigma-Aldrich, Cat# 05030) was added to the cells and incubated for 10 min at RT. Subsequently, the lysed cells were transferred into a 2 mL Eppendorf tube containing 800 µL of digestion mix: 100 µL of 10× NEB1 buffer (New England Biolabs, Cat# B7001S), 100 µL of 10% Triton (Sigma-Aldrich, Cat# T8787), and 600 µL of sterile water. After removing 80 µL of non-digested control, 200 units of HpaII enzyme (New England Biolabs, Cat# R0171L) were added to the digestion mix. Both tubes were then incubated for 3 h at 37 °C with agitation. Following DNA digestion, 80 µL of the digested control were removed and kept on ice with the non-digested control. The remaining digestion mix was centrifuged for 20 min at 16,000 × *g* at 4 °C, and the pellet was re-suspended in 80 µL of sterile water. The 80 µL of digested DNA was then added to the biotinylation mix containing 10 µL of 10× ligation buffer, 1 µL of 10 mM dA/G/T-TP mix, 3 µL of 1 mM biotin-14-dCTP (Jena Biosciences, Cat# E-NU-956-BIO14-S), and 10 units of DNA polymerase I, large (Klenow) fragment (New England Biolabs, Cat# M0210L), and incubated for 1 h at 37 °C with agitation. Following biotinylation, proximity ligation of DNA fragments was performed by adding the ligation mix: 24 µL of 10× ligation buffer (without ATP), 2.4 µL of 100 mM ATP, 2.4 µL of 10 mg/mL BSA (New England Biolabs, Cat# B9000S), and 96 units of of T4 DNA ligase (Thermo Fisher Scientific, Cat# EL0013). Ligation was performed at 25 °C for 3 h with gentle agitation. The crosslink was then reversed by incubating the samples (including the non-digested and digested controls) overnight at 65 °C in the presence of 20 µl of proteinase K (20 mg/mL; Eurobio Scientific, Cat# GEXPROK01-B5), 4 µl of EDTA (500 mM; Sigma-Aldrich, Cat# 03690), and 16 µl of 10% SDS. The next day, DNA was purified by phenol extraction and ethanol precipitation. Briefly, one volume of phenol:chloroform:isoamyl alcohol (25:24:1) (Thermo Fisher Scientific, Cat# 10308293) was added to the samples. Following centrifugation, the upper aqueous phase was transferred to a new 1.5 mL tube, and DNA was precipitated with 1/10 volume of 3 M Na-acetate (pH 5.2) and two and a half volumes of cold ethanol. After incubation at −80 °C for 30 min, the DNA was pelleted, washed once with 70% ethanol, and dried using an Eppendorf concentrator plus speed vac. The DNA was resuspended in 140 µL of 1× TE buffer containing 10 µg/mL RNase A (Thermo Fisher Scientific, Cat# EN0531). The efficiency of the 3C library preparation was evaluated by running aliquots of the 3C libraries, along with non-digested and digested controls, on a 1% agarose gel.

For the Hi-C experiments performed on samples collected after cell sorting, the same protocol was followed, with a fivefold increase in the quantity of each reagent used at every step of the process.

## Chromatin immuno precipitation (ChIP)

Chromatin immunoprecipitation was performed as described[62] with some adjustments. Briefly, frozen cell pellets, previously fixed with 1% formaldehyde (Final concentration), were thawed on ice and re-suspended in 500 µl of lysis buffer I: 20% sucrose (Sigma-Aldrich, Cat# S7903), 10 mM Tris-HCl (pH 8), 50 mM NaCl, and 10 mM EDTA, supplemented with 4 µl of ready-lyse lysozyme. The mixture was incubated at 37 °C for 30 min. Next, 500 µl of lysis buffer II (50 mM Tris-HCl (pH8), 150 mM NaCl, 1 mM EDTA, 1% Triton X-100 and complete protease inhibitor cocktail) was added. The cells were transferred to a 1 mL Covaris tube and sonicated using a Covaris S220 focused Ultra-sonicator. Sonication parameters were set to 210 seconds for EEP samples and 270 seconds for ESP samples (peak power = 140 W; duty factor = 5%; cycles per burst = 200). Following sonication, the samples were centrifuged at 18,000 × g for 30 minutes at 4 °C. The supernatant was transferred into a new 1.5 mL tube, with 50 µL set aside as input and stored at −20 °C until further use. Immunoprecipitation of FLAG-tagged proteins was performed by incubating the cell extracts with 100 µL of ANTI-FLAG M2 affinity gel (Sigma-Aldrich, Cat# A2220) overnight at 4 °C on a rotator. The beads were washed twice with 400 µL of TBS (Tris-HCl 50 mM pH 7.4, 150 mM NaCl) containing 0.05% Tween 20, followed by three washes with 400 µL of cold TBS. Each wash step was performed under rotation for 10 min at 4 °C. The *3xFLAG*-protein-DNA complexes were eluted by adding 2.5 volumes of 3xFLAG peptide solution (prepared as indicated by the manufacturer, Sigma-Aldrich, Cat# F4799) relative to the total packed gel volume. The samples were incubated for 90 min on a rotator at 4 °C, centrifuged for 30 seconds at 5000 × g at 4 °C, and the supernatant was transferred to 1.5 mL LoBind DNA tube. A second elution step was performed similarly. The IP and input samples were treated with RNase A (10 µg/mL) for 1 h at 37 °C and then incubated overnight at 65 °C with proteinase K (final concentration: 50 µg/mL) to reverse the formaldehyde crosslinks. DNA was then purified using a Qiagen MinElute PCR Purification Kit (Qiagen, Cat# 28004) and stored at -20 °C until further processing.

## Processing of libraries for Illumina sequencing

For Hi-C and ChIP libraries preparation for sequencing, samples were first sheared to a target fragment size of 300 bp using a Covaris S220 instrument. Then, sorted Hi-C libraries were prepared as described in ref. 16. For EEP, ESP, anaerobic growth and oxygen shock Hi-C libraries, we used half of the volumes of the indicated reagents in ref. 16.

After sonication of ChIP-seq libraries, IP and input samples were purified using 1.6x volume of AMPure XP beads according to manufacturer instructions (Beckman Coulter, Cat# A63881). DNA was eluted into 40 µL of elution buffer (EB, 10 mM Tris-HCl pH 7.5). After adjusting the volume of the samples to 80 µL with sterile water, DNA fragments were end-repaired by adding 40 µL of the following mix: 12 µL 10× T4 DNA ligase buffer (New England Biolabs, Cat# B0202S), 4 µL 10 mM dNTP mix, 5 µL T4 DNA polymerase (3 U/µL, New England Biolabs, Cat# M0203L), 5 µL T4 Polynucleotide Kinase (10 U/µL, Thermo Fisher Scientific, Cat# EK0032), 1 µL DNA polymerase I, large (Klenow) fragment (5 U/µL, New England Biolabs, Cat# M0210L), and 13 µL H₂O. The reaction was incubated at room temperature (RT) for 30 min, purified using 1.6× AMPure XP beads, and DNA was eluted in 30 µL of EB. Purified DNA was then dA-tailed by adding 5 µL of 10× NEBuffer 2 (New England Biolabs, Cat# B7002S), 10 µL of 1 mM dATP (Sigma-Aldrich, Cat# D4788), 3 µL of Klenow Fragment (3'–5' exo-), (5 U/µL, New England Biolabs, Cat# M0212L), and 2 µL of H₂O. The reaction was incubated at 37 °C for 30 min, followed by heat inactivation at 65 °C for 20 min to inhibit the Klenow Fragment (3'–5' exo-) activity. After purification with 1.6× volume of AMPure XP beads, DNA was eluted in 12 µL of EB. Customized adapters compatible with Illumina sequencing were ligated using the NEBNext Quick Ligation Module (New England Biolabs, Cat# E6056S) for 15 min at RT. The ligation reaction was then purified using AMPure XP beads. First, the sample volume was adjusted to 50 µL with water, and a double size selection step was performed with the beads at a ratio of 0.7×-1.1×. For *3xFLAG*-H-NS and *3xFLAG*-RpoC libraries, PCR amplification was performed in a single 15-cycle reaction using 5 µl of library, 10 µl of 2 µM Illumina primers P1 and P2 mix, 10 µl Phusion 2× High Fidelity Master Mix (New England Biolabs, Cat# M0531S) and 10 µl of H₂O (for both IP and input). For the *3xFLAG*-HilD and *3xFLAG*-H-NS on sorted populations, three PCR reactions were performed for IP samples and one for the input sample. PCR products were purified using Qiagen MinElute PCR purification columns, and primer dimers were removed using AMPure XP beads.

The concentration of Hi-C and ChIP-seq libraries was quantified using a Qubit fluorometer (Invitrogen) with the Qubit dsDNA HS Assay Kit (Thermo Fisher Scientific, Cat# 10606433). Libraries were diluted to 0.5 ng/µL, and their quality was assessed on an Agilent 4150 TapeStation system using Agilent High Sensitivity D5000 ScreenTape (Agilent, Cat# 5067-5592). Libraries were then pooled in equimolar proportions and sequenced as 75 bp paired-end reads on an Illumina NextSeq 500 system or as 150 bp paired-end reads on Illumina NovaSeq X Plus Series. All libraries generated in this study, including the number of replicates, are listed in Supplementary Table 5, Supplementary Table 6 and Supplementary Table 7. The number of replicates per experiment are shown in these tables, indicating in each case when replicates were merged to generate the final data used in the analyses.

## Flow cytometry and fluorescence-activated cell sorting (FACS)

For Hi-C and RNA-seq experiments on sorted subpopulations, the strain carrying the $P_{prgH}$-gfp $Cm^R$ reporter cassette[21] was used (both in WT and hilE mutant), while the hns-3xFLAG $P_{prgH}$-gfp $Cm^R$ strain was applied in ChIP-seq experiments. Cells were grown in LB medium to an OD600 of 2 (ESP) as indicated previously, and then cells were fixed with formaldehyde (2.7% final concentration for Hi-C, 1% final concentration for ChIP-seq). For Hi-C and ChIP-seq, 20 ml of -2 × 10⁸ bacteria/ml cells were fixed as described previously. Fixed cells were pelleted, washed, and stored at -70 °C until sorting.

For RNA-seq, cells were fixed using ice-cold 4% formaldehyde in PBS, following the protocol detailed in ref. 63. Briefly, 1 mL of cells (with ~2 × 10⁹ bacteria) was harvested and resuspended in ice-cold 4% formaldehyde in PBS, followed by incubation on a rotator at 4 °C for 2 h. Fixed cells were washed twice with 1 mL PBS containing 0.01 U/µL SUPERase-In RNAse Inhibitor (Thermo Fisher, Cat# AM2696), and the resulting pellets were stored at -70 °C.

The cell pellets were re-suspended in 20 mL of 1x PBS for sorting, with a MoFlo Astrios^EQ cell sorter (Beckman-coulter). The instrument used a 70-µm nozzle with a sheath pressure at 60 PSI. Home-made PBS (137 mM NaCl Supelco Ref. 1.06104.1000, 2.7 mM KCl Sigma-Aldrich, Cat# P3911, 8 mM Na2HPO4 Ref. PRO-28028.298, 2 mM KH2PO4 Merck Ref. 1.06579.0500) was used as sheath fluid to avoid bactericide or NaN₃ present in commercial sheath liquid. The sorter was calibrated with Ultra Rainbow Calibration particles (Spherotech). Frequency of drop formation was close to 96,000 Hz to reach a flow rate around 50,000 events per second. A first gate was made on a Forward Scatter (FSC-Height) - Side Scatter (SSC-Height) dot plot to select the bacterial population. Doublets were discarded using an SSC-Area – SSC-Height dot plot. Finally, the cut-off of GFP positive cells was determined by comparing the $P_{prgH}$-gfp sample to a control bacteria sample, using a 525/52-nm bandpass filter for collecting fluorescence emission signal. Sorting, according to green fluorescence intensity, was made under constant cooling to 4 °C of both the input chamber and the collection tube holder. The collection tubes (50 mL Falcon) were treated with the Sigmacote® (Sigma-Aldrich, Cat#SL2) prior to the sorting process. On average -2 × 10⁷ bacteria were collected per fraction (2–3 fractions of 50 ml per sorted population). For the WT Hi-C experiment, sorting required 30 h to accumulate -2 × 10⁸ total bacteria. In contrast, for the ΔhilE strain, given that -20% of the cells were GFP + , sorting took only

15 h to collect the same number of total bacteria. For the ChIP-seq experiment, each replicate took 24 h to collect a total of ~$5 \times 10^8$ bacteria, while RNA-seq sorting was completed in about 6 h per replicate, yielding ~$4 \times 10^7$ bacteria in total. After sorting, bovine serum albumin (BSA, Sigma-Aldrich, Cat# A2153) was added to the cells to a final concentration of 0.2%, in order to make centrifugation more efficient. Cells were then collected by centrifugation at 2599 x g for 50 minutes at 4 °C. Of note, the cell pellet was often barely visible. After centrifugation, the supernatant was quickly removed, and the pellets were stored at -70 °C until further processing.

## Isolation of Total RNA

RNA extraction was performed as described by ref. 64 with some modifications. Frozen pellets of sorted cells were thawed on ice and resuspended in 1 mL of TRIzol (Qiagen, Cat# 79306), followed by the addition of 400 µL of chloroform. After a 3-minute incubation, the mixture was centrifuged at 20,000 × g for 15 minutes at 4 °C. The upper aqueous phase containing RNA was carefully transferred to a new tube, and 600 µL of isopropanol was added to precipitate the RNA at room temperature. Following a 30-minute centrifugation at 20,000 × g at 4 °C, the supernatant was discarded. The RNA pellet was washed with 500 µL of 70% ethanol, air-dried, and resuspended in RNase-free water by shaking at 900 rpm for 5 minutes. RNA was treated with DNase according to the manufacturer's instructions (Invitrogen™ TURBO DNA-free™ Kit, Cat# AM1907). RNA concentrations were determined using either a Nanodrop spectrophotometer or a Qubit fluorometer (Invitrogen). RNA quality was assessed with an Agilent 2100 Bioanalyzer, and samples were stored at −20 °C.

## RNA-seq library preparation

RNA-seq libraries were prepared using the Illumina Stranded Total RNA Prep kit. The standard protocol provided with the kit was followed, with a number of modifications; (1) 0.5 x reaction volumes were used, (2) for ribosomal RNA (rRNA) depletion, the microbiome rRNA depletion mix (DPM) was added to the Depletion Probe pool (DP1) to the same final amount (0.5 µL both), (3) fragmentation was done for 1 min at 94 °C instead of 2 min, (4) for cleanup of ligated fragments, 1 x volumes of AMPure beads were used instead of 0.8 x.

## Structured Illumination Microscopy (SIM) Imaging

The *prgH(stop)mcherry parS*$^{pMT1}$ harboring *pSPB5* cells were grown to OD600 = 2.0 (ESP) as previously indicated, and at least $1 \times 10^8$ bacteria were fixed with formaldehyde-glutaraldehyde solution for 20 min on a rotating wheel at 4 °C, and subsequently stained with Hoechst 33258 solution (a.k.a Hoechst) for DNA visualization.

For SIM imaging 1 µl of cell culture was spotted on a 1.5% (w/v) agar pad covered with coverslips. Structured Illumination Microscopy imaging was performed on a Lattice SIM Elyra 7 (Zeiss) equipped with a Plan-Apochromat 63x oil immersion objective (numerical aperture = 1.4), using a quad band filter block (405/488/561/642) for all the channels, coupled with a PCO.edge 4.2 sCMOS camera (Execelitas Technologies) and driven by ZEN 3.0 SR FP2 software (black v.1.60, Zeiss). First, to optimize the quantification of the faint red signals, the z-stack of the mCherry channel was acquired in wide-field with 5% of 561 nm laser, a z-step of 0.5 µm and an exposure time of 1 sec. For both Hoechst and YFP imaging we used an illumination pattern of 27.5 µm to structure the illumination of the 405 nm (50 mW, 3 % power, 500 ms of exposure per phase) and 488 nm laser (100 mW, 3 % power, 500 ms of exposure per phase). Z stacks of the whole bacteria, with a step of 0.091 µm, were acquired by switching between both channels at each Z to ensure optimal matching between the wavelengths. To reconstruct the 3D-SIM data we used the SIM² algorithm (Zeiss) with the following parameters: Hoechst (5 iterations, Regularization weight of 0.1, processing sampling 4, output sampling 4) and YFP (20 iterations, Regularization weight of 0.03, processing sampling 4, output sampling 4).

To prevent chromatic aberrations, we used 200 nm fluorescent beads to generate a channel alignment matrix to be applied to the processed data.

## SIM data analysis

A dedicated analysis workflow was set up to compare the relative location of SPI-1 foci within the nucleoid (see Supplementary Fig. 8 for details). The first steps of image correction and correlation of wide-field and SIM² images were performed using custom scripts in FIJI (REF N66). The data were then processed in IMARIS (Bitplane, v. 9.4) for all 3D quantification steps. Once the data had been extracted, Jupyter Notebooks (Python 3.9) were used to determine the thresholds, perform the statistical tests and generate the figures. For SPI-1 localization, two independent experiments were combined and analyzed together, whereas a single replicate was analyzed for *yqgF* localization.

## Processing of sequencing data

Sequencing data, including Hi-C, ChIP-seq and RNAseq[34], were processed starting with adapter trimming using Cutadapt. Paired-end reads were then aligned to the *Salmonella enterica* serovar Typhimurium strain SL1344 genome (GenBank: FQ312003.1) using Bowtie2 (v 2.4.2)[65]. The mapping and analyses were initially performed with both chromosomal and plasmid sequences included in the FASTA file. However, for simplicity, plasmids were excluded from the final results, and the analysis focused exclusively on the chromosomal sequence.

## Hi-C data processing

Hi-C data were further analyzed using the OCHA pipeline in R, as described in ref. 66. Briefly, mapped reads were converted into.pairs files, which were subsequently filtered to remove invalid interactions and PCR duplicates. The filtered data were binned into sparse contact matrices at multiple resolutions, and normalization was performed using the Balanced normalization method to correct for experimental biases. The resulting contact matrices were imported into R as a HiCExperiment object for further analysis, including the generation of multi-resolution contact matrices. For visualization, contact maps were displayed as $\log_{10}$ transformed matrices using MATLAB (2015). The scale bar next to the maps represents the contact frequencies in $\log_{10}$ the darker the color, the higher the frequency of contacts between given loci. Custom MATLAB scripts were used to plot the Hi-C matrices, using a Gaussian filter (H = 1) for global view of the chromosomes but with no filter for local analyses at SPIs. The binning of the contact maps displayed in figures and used for analyses is 1 kb.

## ChIP-seq data processing

For the ChIP-seq analysis, after mapping paired-end reads were then merged and sorted with Samtools, and PCR duplicates were removed using the Picard MarkDuplicates tool. Subsequently, SAM files were converted to BAM files, indexed, and processed to generate bedGraph and bigWig files using bamCompared from DeepTools2[67]. This tool allowed us to compare the ChIP-seq data to the INPUT and normalize it in Bins Per Million mapped reads (BPM). For Hi-C correlations and determination of protein occupancy (H-NS and RNA polymerase), 1 kb bins were used. No binning was used for the rest of analyses.

For ChIP targeting HilD and H-NS, two replicates were merged, while for RNA polymerase, four replicates were merged at the BAM file level. The resulting bigWig files were visualized using Integrative Genome Viewer (IGV), and bedGraph files were visualized using MATLAB (2015) and used for correlations. Peak calling was performed with MACS2 callpeak[40].

## RNA-seq data processing

For RNA-seq analysis, after alignment, SAM files were sorted, converted to BAM format, and indexed. BAM files were subsequently converted to WIG format using bamCoverage or Galaxy[68] with counts

per million (CPM) normalization. Read quantification in sense and antisense orientations was performed using featureCounts program (v 2.0.1)[69].

Differential gene expression analysis was conducted in R using the SARTools pipeline (Statistical Analysis of RNA-Seq data Tools, v1.8.1), which is based on DESeq2[70]. Quality controls, raw count normalization, and identification of differentially expressed genes were systematically performed. Principal Component Analysis (PCA) and hierarchical clustering were applied to data transformed using Variance Stabilizing Transformation (VST) to ensure homoscedasticity. The Benjamini-Hochberg method was used for *p*-value adjustment with a statistical significance threshold of 0.05. The condition "GFP negative" was defined as the reference for comparisons. Genes with null counts across all samples and predefined low-quality features were excluded prior to analysis. Genes that were differentially expressed in the sorted populations, were identified by considering those whose transcriptomic changes were associated with at least a three-fold expression difference (Supplementary Data 1). The statistical report of this RNA-seq analysis is presented in Supplementary Data 5. Additionally, to facilitate correlations with Hi-C data, DESeq2-normalized counts per kilobase for each gene under all experimental conditions were determined (Supplementary Data 6).

To classify gene expression levels, the normalized counts generated using the SARTools DESeq2 pipeline were further adjusted for gene length (calculated as DESeq2 normalized reads divided by gene size and per million reads, transcripts per million (TPM)). This allowed us to compare directly the relative levels of gene expression of individual genes. Expression levels were classified from poorly expressed to highly expressed as previously done in ref. [71]. Specifically, poorly expressed genes are defined as those whose expression levels fall within the 25th percentile of the expression distribution across the entire chromosome. Highly expressed genes (HEGs) are defined as those whose expression levels lie between the third quartile (Q3) and the maximum observed expression value across the chromosome'

The data analysis was conducted using R software, and visualization of RNA-seq results alongside genomic annotations was performed using the IGV[72]. Relevant genes were attributed to a KEGG orthology category based on their annotation in https://www.genome.jp/kegg-bin/show_organism?org=sey.

Regarding the comparative analysis of our transcriptomic data with the previously published dataset from[34] on the 4/74 strain, the published data were reanalyzed using the same pipeline applied to our dataset. Both datasets were mapped to the SL1344 genome, as the two strains are closely related. Specifically, *S*. Typhimurium 4/74 is the parental strain of SL1344, this latter strain has a *hisG46* mutation[57]. Therefore, the two strains differ by only eight SNPs, with 4/74 being a prototroph possessing a functional histidine biosynthesis pathway[34]. Key metrics, such as log$_2$ fold change and adjusted *p*-values, were calculated to identify differentially expressed genes in both datasets. To visualize the agreement or divergence between the datasets, a comparative scatter plot was generated in R, including interactive features (*e.g.*, Plotly) to explore the expression patterns of specific genes in more detail.

### Comparison of genomic data (Z-transformation)

We performed correlation analysis as previously done in ref. [15]. Briefly, Z-transformation allows the standardization of data and makes two samples with different units comparable. It is based in computing the deviation of a signal with respect to its mean value. It centers the mean of the distribution of the signal on 0, and transforms each value of the distribution into a standard score (or Z-score). A Z score corresponds to the number of standard deviations by which an observed value differs from the mean value of the entire dataset. $z_i = (x_i - m)/ s$, where $x_i$ is the original value in the sample, $m$ is the mean of the population and $s$ is the standard deviation of the population. Z-scores are convenient to compare and visualize different signals of various strengths.

Here, we computed the Z-scores of H-NS ChIP-seq (log2 ChIP/INPUT) with Hi-C contact signals binned at 1 kb to compare them with each other. To correlate H-NS occupancy with AT content, we first determine the Adenine and Thymidine (AT) content (%) in SPI-1 using a sliding window of 50 bp with a step size of 50 bp. We then computed the Z-score of these data and correlate it with the Z-score of H-NS ChIP-seq (log2 ChIP/INPUT) binned at 50 bp.

### Determination of SPI-1 AT content

To determine the percentage of AT along SPI-1 used in Fig. 4, we used a sliding window of 200 bp in size with an increasing step of size of 1 bp. For correlation with the H-NS occupancy, AT content was computed using a sliding window of 50 bp in size with an increasing step of 50 bp.

### Frontier index analysis

To identify boundaries we applied the Frontier Index (FI) method as described in detail in ref. [35] and summarized in Hi-C contact maps that can occur at any scale, meaning that the method is useful to identify domain organization irrespective of the underlying scale. Specifically, at a given locus (let us say *i*), two frontiers can be found at most, each one corresponding to a side of the locus—one frontier indicates that the neighboring loci on the same side of the locus *i* tend to make more contacts with each other with respect to neighboring loci separated by the same genomic distances but located on either side of the locus. Visually, identified frontiers can be vertical (or upstream, green peak) or horizontal (downstream, orange peak).

### Ratio of Hi-C contact matrices

To qualitatively compare normalized contact maps we took a ratio of the matrices. This ratio is computed for each bin of the map by dividing the amount of contacts it presents in one condition by the amount of contacts in the other condition, and applying a log2 transformation to the resulting matrix. The log2 of the ratio is plotted with a Gaussian filter (H = 1). The color code reflects a decrease or increase of contacts in one condition compared to the other (blue or red signal, respectively). No change is represented by a white signal.

### Reporting summary

Further information on research design is available in the Nature Portfolio Reporting Summary linked to this article.

## Data availability

The ChIP-seq, RNA-seq and Hi-C data generated in this study have been deposited in the NCBI Gene Expression Omnibus database under accession code GSE289504, including normalized contact matrices, BAM files for RNA-seq, bigwig files for ChIP-seq. The ChIP-seq and RNA-seq data generated in this study are provided in Supplementary Data 1 and Supplementary Data 2. In addition, long-range contact analyses, Frontier Index analyses are provided as Source Data file. RNA-seq data from[34] was used for comparative analyses and are available in the NCBI Gene Expression Omnibus database under accession number GSE49829. Source data are provided with this paper.

## Code availability

The scripts used for data analyses are available on the following Github links: RNA-seq, analyses: https://github.com/PF2-pasteur-fr/SARTools, Hi-C analyses: https://github.com/js2264/OHCA, Hi-C frontier index analyses: https://github.com/VickyTche/Frontier_Index_Salmonella, 3D-SIM analyses: https://github.com/Rom-LB/SPI1 and ChIP-seq: https://deeptools.readthedocs.io/en/latest/index.html.

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

## Acknowledgements

We thank the members of the FunGBugs consortium (Olga Soutourina, Emanuele Biondi, Peter Mergaert and Céline Hernandez) for fruitful discussions and to the I2BC Scientific Council for funding the FunG-Bugs transversal project. We also want to thank to Pascaline Tirand and Fanny Culot for daily help and preparing culture media, to Alicia Nevers for helping us with R scripts to analyze the transcriptomic data, to Sokrich Ponndara for constructing pSPB5, Pierre Grognet for advice on the analysis of ChIP-seq data using Deeptools and to Melina Gallopin, for useful advice on statistics. We finally want to thank all the Boccard team members, past and present, for useful discussions. This work was supported by the French National Research Agency: grant number ANR-20-CE35-005 (V.S.L), by the 80 Prime CNRS project SIRIG (V.S.L.) through the MITI interdisciplinary exploratory research programs. We acknowledge the sequencing and bioinformatics expertise of the I2BC High-throughput sequencing facility, supported by France Génomique (ANR-10-INBS-09). The present work has benefited from Imagerie-Gif Flow cytometry and light microscopy facilities supported by l'Agence Nationale de la Recherche (ANR-24-INBS-0005 FBI (BIO-GEN); ANR-11-IDEX-0003-02/ Saclay Plant Sciences). Finally, we acknowledge the use of OpenAI's ChatGPT to improve the scripts used to analyze data.

## Author contributions

M.K. performed all genomic experiments, strain construction, FACS, and fluorescence microscopy, analyzed data, wrote and revised the paper. M.B., performed cell sorting experiments analyzed data and wrote and revised the paper; R.L.B., performed 3D-SIM experiments, developed analysis tools and wrote and revised the paper; E.V.D., prepared and sequenced low input RNA-seq libraries of sorted cells, wrote and revised the paper; C.J.D. helped devise the project, assess the data and wrote and revised the paper; S.B.M., analyzed data, wrote and revised the manuscript and conceptualized the work; F.B., wrote and revised the manuscript and conceptualized the work; V.S.L., devised, conceptualized and supervised the work, acquired funding, performed analyses, wrote the original draft and revised the manuscript.

## Competing interests

The authors declare no competing interests.
