## [Transparent Peer Review file · Nature Communications]

Bacterial chromatin remodeling associated with transcription-induced domains at pathogenicity islands

Corresponding Author: Dr Virginia Lioy

Version 0:

Reviewer comments:

Reviewer #1

(Remarks to the Author)

This study by Kortebi et al., investigates the regulation of a key pathogenicity island SPI-1 in *Salmonella Typhimurium*, which encodes for Type III Secretion System (T3SS) essential for host infection. However, the stochastic and bistable expression of this pathogenicity island makes it complicated to dissect the molecular basis of its activation. Here the authors have exploited the bistable expression of this island in stationary phase to sort SPI-1ON and SPI-1OFF cells to assess the differences between the two populations. By integrating Hi-C, ChIP-Seq and RNA Seq across these populations, and different growth conditions and regulatory mutants, they show local chromatin remodeling in the form of transcription induced domains (TIDs) in SPI-1ON cells. Such remodeling which is dependent on HilD (transcriptional regulator of the island) is associated with increased transcription and reduced occupancy of the nucleoid associated protein (NAP) H-NS. Further they show that the chromosomal region harboring the pathogenicity island is localized to the periphery of the nucleoid in cells where SPI-1 is turned on.

By adopting a clever strategy to distinguish cells with and without SPI-1 activation, and using systematic functional genomics approaches, the authors have clearly delineated local changes in chromosome organization and NAP binding at high resolution, which are associated with transcriptional activation of a pathogenicity island. Overall, the study provides novel and important insights into regulation of silent genetic elements that needs to be activated in specialized contexts. The authors should consider the following comments to strengthen their findings prior to publication:

Main comments:

1. Fig. 2F. provides important evidence for the role of HilD in inducing the chromatin-level changes at the SPI-1 locus. However, this experiment was carried out in a different condition (ESP) than the rest of the experiments in this figure (which were done in GFP+ cells). Authors should provide evidence that GFP+ cells are not obtained in hilD deletion background, and should also include H-NS/ RNA-Pol ChIP for hilD deleted cells in the ESP condition.
2. Authors implicate a role for active transcription in the chromatin remodeling. However, a direct assessment of the same is lacking. One method is to treat GFP- and GFP+ cells with rifampicin after the FACS sorting, and carrying out Hi-C to probe more directly a role for transcription. Along with Hi-C, ChIP for H-NS, HilD and RNA polymerase can also be conducted to further strengthen the observations.
3. What role is H-NS playing in the remodeling? To support the conclusions made by the authors (including the proposed model) H-NS would have to be deleted and Hi-C should be conducted in this background. If H-NS is essential, authors can consider depleting the protein, or combining the H-NS deletion with rpoS mutants that provide a partial rescue to the H-NS null phenotype.
4. Section on spurious transcription from H-NS silenced SPI chromatin lacks clarity and experimental strength. It is clear in Supplementary Figure 5 that genes occupied by H-NS has lower sense and antisense transcription. But to identify transcriptional changes specific to pathogenicity islands, sense and anti-sense activity of HNS-bound pathogenicity islands should be compared to the rest of HNS-bound genes and HNS-free genes. This will clarify if the spurious antisense transcription observed here is unique to these pathogenicity islands. For this comparison genes from all pathogenicity islands could be pooled to avoid clusters with very low gene numbers. For example, in Figure 3, the comparisons are made between clusters ranging from 3 genes to >4000 genes. In addition, the logic for using highly expressed genes is vague. If the authors maintain this cluster, the logic should be stated in text and how HEGs are defined should be clearly stated.
5. The significance of the SPI-I locus mobility to nucleoid periphery is unclear. How do the authors propose this occurs? Authors should also provide imaging of a control region not within the SPI-1 locus to show that this observation is SPI-1 specific. Additionally, this mobility should increase in hilE deletion and be abrogated in hilD deletion.

6. The model that the authors have proposed is speculative. A particular observation that needs further validation is the role of Spot-42 srRNA, which is brought up recurrently in the manuscript as being responsible for HlID positive regulation. In addition, the upregulation of this Spot-42 gene spf contrasts with previous studies, and hence it would be a key experiment to test the importance of this upregulation. Given the model proposed by the authors, expression of Spot-42 in exponential phase would induce SPI-1 bistable expression and absence of the srRNA could lead to lack of such bistable expression in stationary phase. More interestingly what could be the impact of depleting this srRNA on formation of TIDs. Without further experimental evidences, it is recommended that the authors tone down the speculative nature of the model.

Minor comments:

1. Order of figures should match their citation in the text.
2. Please label the conditions for each figure panel in the figures. Presently, it is very confusing to know which panel corresponds to which experimental condition.
3. Please provide information on the number of replicates for each experiment.
4. It would be interesting to see if there is a correlation between mCherry intensity (indicating the extent of SPI-1 activation) and the distance of the genetic locus?
5. L45: "de-repressed" instead of "depressed"
6. L98: You should elaborate on the 39 genes - how many of them are SPI-1 and non-SPI-1 and what proportion of SPI-1 genes have been identified as upregulated.
7. L128: Change 'observed' to 'observe'
8. L144: The significance of the observation about TIDs encompassing hIIA and invF needs to be stated here. For example, no change in H-NS occupancy is observed. Could you comment on this?
9. L205-210. RNA polymerase itself could be a source of H-NS dissociation from these loci
10. Fig. 4C-D. It is unclear what the reader is supposed to focus on. Could you please indicate via arrows/ lines?
11. L220: There is no Figure 4F as cited in text.
12. L234: It is unclear what you mean by 'assure a source of repressor'
13. L257-259 is speculative and should be removed unless data to support this conclusion is provided.
14. L365: The plasmid pGBD3.2 is not listed in the plasmid sheet
15. Construction of pMKB1 and pSPB5 should be included.
16. L489: Spelling of cell pellets needs correction
17. Fig. 3G legend: Spelling of beginning needs correction
18. Fig. 4A: Show a representative 1 kb region with H-NS occupancy and contact signal outside SPI-1
19. Supplementary Figure 5 legends - Clarify what Category 1 is.

Reviewer #2

(Remarks to the Author)

Reviewer #3

(Remarks to the Author)

In this paper Kordei et al use a combination of global and microscopic methods to study H-NS and transcriptional silencing in Salmonella. The experiments are done to a high standard, and I have no issue with anything in results sections 1, 2 or 5. However, I found the way the story is presented in results sections 3 and 4 quite confusing and I wonder if some of the data can really be interpreted as described. My comments are quite lengthy because I'm trying to address some important but subtle points. I hope the review is clear and useful to the authors.

SECTION BEGINNING LINE 159: I found the way this section was presented to be very confusing. I think this is largely because of the way the rationale for the experiments is set up. The paragraph starts by suggesting a discrepancy between existing stories in the literature (i.e. H-NS repressing pervasive transcription vs some pervasive transcription being evident at H-NS bound loci). My personal interpretation is that the two stories are consistent with each other; H-NS deletion permits high-level spurious transcription, but some is still detectable, albeit at much lower levels, when H-NS is present. My recollection is that both papers, and several more recent studies, show data consistent with this. There's quite a lot to pull apart here so I've tried to separate the issues below.

1. Assuming there is some sort of disagreement, which I don't think there is, the authors then describe experiments that don't really discriminate between the two possibilities. Instead, they compare levels of spurious transcription within H-NS bound regions, and other parts of the chromosome, in GFP- cells. The conclusion seems to be that H-NS represses spurious transcription but that these levels of spurious transcription are higher than expected for some H-NS bound SPIs*. I'm not suggesting the authors do it but, if they really think there is some sort of disagreement in the literature, surely the key experiments would be to: i) test H-NS +/- cells to determine effects on spurious transcription or ii) do the experiment in GFP+ cells to determine if levels of spurious transcription increases when the SPIs are turned on (a caveat is that I don't think RNA-seq easily measures spurious transcription, see below).

(*to my mind the key question is whether this transcription is truly occurring when H-NS is still bound, implying some sort of chromatin remodeling, or whether H-NS is being stochastically released, and/or the SPIs turned on, within the population of GFP- cells. I also wonder if these SPIs are unusually AT-rich, compared to other H-NS bound loci, which might account for

the higher spurious transcription)

2. Lines 174-178: "Altogether, these results confirm that high H-NS occupancy co-exists with spurious sense and antisense transcriptional activity at repressed SPI-1 and other silent SPIs (Figure 3, blue boxplots). This is further supported by the low RNA polymerase occupancy within repressed SPIs observed in non-sorted exponentially growing *Salmonella* cells (Figure Supplementary 6)." I don't follow the authors logic here, why does the low RNAP occupancy support the presence of spurious transcription?

3. I think there may be differences between definitions of transcriptional events used here and in prior papers. There are also differences in exactly what is measured by different experimental tools in the various papers. In this work, the terms "pervasive" and "spurious" transcription are used interchangeably, unless I misunderstand. Obviously, there's no formal definition, but to me the meanings are subtly different. I would describe pervasive transcription as a "catch all" term referring to the observation that RNAs are made everywhere, and on both strands, to some extent. There are two main causes of this (transcription initiation in unusual locations and inefficient termination of mRNAs). Spurious transcription, at least as I see it, refers to the sub-category of initiation in unusually locations and, more specifically, the propensity of RNAP to initiate from many sites in AT-rich DNA. With respect to exactly what is measured, RNA-seq will find reasonably abundant RNAs and detect some RNAs resulting from spurious transcription. However, an issue is that spurious sense transcripts, from within genes, can be hidden by the overlapping signal from the full-length mRNA. Hence, TSS mapping also allows for much easier differentiation between mRNAs and spurious intragenic sense RNAs. A further issue is that spurious RNA's get terminated quickly by Rho. Again, this means spurious RNAs are easier to find if only the TSS is mapped, instead sequencing total RNA.

4. To summarize the above, I think all existing data are consistent with H-NS repressing spurious transcription, but not to the point where it becomes undetectable. If the authors agree that this is what the existing data tell us, it sounds like the interesting result here is that some SPIs have unusually high levels of background spurious transcription, when H-NS is present. I think the section would make a lot more sense if presented in this way.

SECTION BEGINNING ON LINE 184: I think this is the key question the authors skirt around in the section discussed above; does transcription (spurious or not) evict H-NS from the DNA? I think this is a very difficult question to answer unequivocally and would require a single cell approach. Presumably, the populations of GFP- and GFP+ cells collected are not truly homogeneous (i.e. GFP- and + interconvert, at low levels, all of the time).

1. The authors show that a silent SPI-1 exhibits low level spurious transcription and high-level H-NS binding. I think there are several plausible explanations for this but all fall into one of two broad models: i) RNAP can make spurious RNAs at low levels when H-NS is bound, perhaps implying local chromatin rearrangements or ii) in a small number of GFP- cells, H-NS is transiently released from the DNA and transcription occurs. At the population level, both models would be consistent with high H-NS binding but low transcription. I'm not sure the approach here can differentiate between the two possibilities.

2. The authors show that full induction of SPI-1 (in GFP+ cells) leads to reduced (but not abolished, see point 3 below) H-NS binding. The implication is that transcription does evict H-NS, which I can believe. However, it's difficult to know what gives rise to the residual H-NS binding signals. Does H-NS truly remain bound? Does this signal come from a low background of cells that have reverted to the GFP- phenotype? Does H-NS transiently release the DNA and rebind in such a way that can't be detected at the level of the whole GFP+ population?

3. I don't agree with the statement that "H-NS-free regions were found all along the central region of SPI-1, the largest ones at the *hilA* and *invF* regulon, specifically within the genes *orgB*, *orgA*, *prgK*, *prgJ*, part of *prgH*, *invG*". I would say that H-NS binding is reduced but remains higher than background levels since elsewhere in Figure 4D.

OVERALL THOUGHTS

This is a very nice piece of work; the results in Figures 1 and 2 are great and reveal expected results. i.e. increased transcription generates TIDs and H-NS bound regions have very low levels of long-distance contacts (indicating bridging). Similarly, movement of transcribed SPI-1 to the periphery of the nucleoid is nicely demonstrated in Figure 5, again as expected. With respect to what is truly new/unexpected, the major advance would be to determine if H-NS is fully evicted from transcribed regions or remains associated in some way. In this respect, I don't think the data presented can provide a complete answer. Whilst the model in figure 6 is very plausible, there are other interpretations. It's also notable that the model isn't particularly different to that shown in figure 6 of Figueroa-Bossi et al. That said, the authors do improve on the Figueroa-Bossi paper by better separating out the H-NS binding pattern in SPI-1 induced vs uninduced cells.

Minor comments:

Lines 43-46: Is it worth mentioning temperature/osmolarity?

PMID: 33245158 is probably worth a mention regarding H-NS silencing of intragenic promoters.

Title and various places in the paper: I think it would be helpful to define what is meant by remodeling. To me, this means that the chromatin remains largely intact but is altered. This is a personal opinion, but if H-NS is completely evicted by transcription I would not define this as remodeling.

Line 280: "We did not observe specific bridging between or within repressed SPI loci". Presumably it's impossible to say, at

this resolution, if you really see short range bridging (i.e. between H-NS molecules bound to DNA sites separated by less than ~2 kb). Worth mentioning?

Version 1:

Reviewer comments:

Reviewer #1

(Remarks to the Author)

Overall, authors have presented a thorough revision that addresses most concerns previously raised. I congratulate the authors on this excellent work. Only a couple of minor typographical errors are highlighted below:

1. L99 - There are only 3 SPI-4 genes as per Supplementary Table 2.
2. In Supplementary Figure 5, there is no g panel, while it is indicated in legends.

Reviewer #2

(Remarks to the Author)

Reviewer #3

(Remarks to the Author)

The authors have reworked earlier parts of the results section to make the text clearer and better aligned with the existing literature. I think the changes improve the text and make the story clearer. The changes include better defining spurious vs pervasive transcription and a better explanation of existing stories describing interplay between H-NS and RNAP in coding sequences.

I'm not personally convinced by the description of some SPI regions as H-NS free, but, as the authors describe how they arrive at this description, and readers can easily make up their own minds, I don't feel particularly strongly about this. Whether the authors observe no or low levels H-NS binding doesn't really impact the overall story. Happy to agree to disagree.

With respect to my prior comment that "H-NS bound regions have very low levels of long-distance contacts (indicating bridging)" apologies, this was worded this clumsily. I was trying to say that H-NS bound regions don't show high levels of long-range contacts and that long range contacts indicate bridging. I agree with the authors' interpretation of the data.

Overall, the paper is improved and, as on the first occasion, I enjoyed reading it. I'm still left with the feeling that the results are not unexpected, and that there is only a modest advance on what we already know, but that is not a comment on the quality of the work or manuscript. The data support the conclusions and the text puts the work in context of the existing literature.

Ref: *Nature Communications* manuscript NCOMMS-25-20136-T

Manuscript: Bacterial chromatin remodeling associated with transcription-induced domains at pathogenicity islands

Authors: Kortebi et al

Revision date: Octobre 2025

Corresponding author: Virginia S. Lioy

Point-to-Point response to the reviewers.

We want to thank the reviewers for their valuable comments on our manuscript. We have performed additional analysis and experiments based on their feedback. All these new results are included in the revised version of the article. A detailed response to all the comments can be found below

REVIEWER COMMENTS

Reviewer #1 (Remarks to the Author):

This study by Kortebi et al., investigates the regulation of a key pathogenicity island SPI-1 in *Salmonella* Typhimurium, which encodes for Type III Secretion System (T3SS) essential for host infection. However, the stochastic and bistable expression of this pathogenicity island makes it complicated to dissect the molecular basis of its activation. Here the authors have exploited the bistable expression of this island in stationary phase to sort SPI-1ON and SPI-1OFF cells to assess the differences between the two populations. By integrating Hi-C, ChIP-Seq and RNA Seq across these populations, and different growth conditions and regulatory mutants, they show local chromatin remodeling in the form of transcription induced domains (TIDs) in SPI-1 ON cells. Such remodeling which is dependent on HilD (transcriptional regulator of the island) is associated with increased transcription and reduced occupancy of the nucleoid associated protein (NAP) H-NS. Further they show that the chromosomal region harboring the pathogenicity island is localized to the periphery of the nucleoid in cells where SPI-1 is turned on.

By adopting a clever strategy to distinguish cells with and without SPI-1 activation, and using systematic functional genomics approaches, the authors have clearly delineated local changes in chromosome organization and NAP binding at high resolution, which are associated with transcriptional activation of a pathogenicity island. Overall, the study provides novel and important insights into regulation of silent genetic elements that needs to be activated in specialized contexts.

We thank the reviewer for these positive remarks.

The authors should consider the following comments to strengthen their findings prior to publication:

Main comments:

1. Fig. 2F. provides important evidence for the role of HilD in inducing the chromatin-level changes at the SPI-1 locus. However, this experiment was carried out in a different condition (ESP) than the rest of the experiments in this figure (which were done in GFP+ cells). Authors should provide evidence that GFP+ cells are not obtained in hilD deletion background, and should also include H-NS/ RNA-Pol ChIP for hilD deleted cells in the ESP condition.

All experiments shown in Figure 2 were carried out in early stationary phase (ESP), the only difference between samples is that in WT cells, experiments were performed in sorted populations (GFP⁻ & GFP⁺ cells), while they were performed in non-sorted populations in the *hilD* mutant strain. In the *hilD* mutant strain the number of GFP⁺ cells was negligible (0.3%±0.2, see Table 1). **This result is not surprising since the role of HilD as a master activator of SPI-1 is well-established** (Saini *et al.*, 2010). **This information was already included in Supplementary Table 1 in the original version of the article.** To avoid confusion, in the revised version, this table has been moved to the main text (new Table 1, see below). We have performed a new replicate for the quantification of GFP⁺ cells in this strain, allowing us to present the mean from three independent experiments (Table 1).

Table 1. Percentage of GFP⁺ cells in the different strains and conditions studied, as determined by FACS

Strain	% GFP+ cells	
	EEP (OD 0.1)	ESP (OD 2)
P_{prgH}-gfp	0.7±0.5	10.7±1.1
hilD P_{prgH}-gfp	0.10±0.1	0.3±0.2
hilE P_{prgH}-gfp	1.6±1.1	29±4

Concerning the ChIP-seq experiments, in the submitted version of the article, we showed in Supplementary Figure 3 the binding occupancy of H-NS, RNA polymerase and HilD in early exponential phase (EEP) within SPI-1. In EEP the level of GFP⁺ cells are similar to that of a *hilD* deletion mutant strain in ESP (0.7±0.5 vs 0.3±0.2).

In summary, given the well-established role of HilD as master activator of SPI-1 (Saini *et al.*, 2010) and the analyses presented in the revised version—including the number of GFP⁺ cells in the absence of *hilD* in ESP and the ChIP-seq of H-NS and RNA polymerase in repressed conditions (EEP) that were included in the original version of the article, **we provide strong evidence supporting the role of HilD in promoting conformational changes in active SPI-1 chromatin.**

2. Authors implicate a role for active transcription in the chromatin remodeling. However, a direct assessment of the same is lacking. One method is to treat GFP⁻ and GFP⁺ cells with rifampicin after the FACS sorting, and carrying out Hi-C to probe more directly a role for transcription. Along with Hi-C, ChIP for H-NS, HilD and RNA polymerase can also be conducted to further strengthen the observations.

Unfortunately, it is not possible to treat GFP⁻ and GFP⁺ cells with rifampicin post-sorting since cells are fixed with formaldehyde prior to cell sorting. This is a key step and cannot be omitted, as cells are sorted over several days prior to Hi-C or ChIP-seq. Therefore, post-sorting treatments are not feasible. In addition, rifampicin treatment was extensively used in several bacterial models when studying chromosome conformation using low resolution contact maps, and therefore it is a well characterized phenomenon (Le *et al.*, 2013; Le and Laub, 2016; Ponnadara *et al.*, 2024). It was recently used to characterize the role of active transcription, RNA polymerase binding and positive supercoiling in TIDs formation, and consequently, in chromatin remodeling **in non-physiological conditions in *E. coli*** (Bignaud *et al.*, 2024). In our manuscript, we took advantage of the regulatory network of SPI-1 to study silent chromatin and active chromatin, **in physiological conditions.** Rifampicin treatment will

definitely impact other loci and the local structure of the chromosome globally. Our work stands out by presenting original, physiologically relevant data that advance our understanding of SPI-1 regulation. We provide solid data supporting the role of transcription in chromatin remodeling in SPI-1, such as transcriptomic data obtained in sorted populations, in which we observed more than 3-fold upregulation of SPI-1 genes (Figure 1, main text), and the quantification of a well characterized single copy transcriptional reporter that reveals the transcriptional activation of SPI-1 (Hautefort *et al.*, 2003).

In conclusion, we respectfully disagree with this reviewer, since as mentioned above the role of transcription in TID formation is very well characterized. Furthermore, the Hi-C and ChIP-seq experiments asked in the absence of transcriptional activation of SPI-1 in **physiological conditions** were already present in the original version of the manuscript in Supplementary Figure 3.

3. What role is H-NS playing in the remodeling? To support the conclusions made by the authors (including the proposed model) H-NS would have to be deleted and Hi-C should be conducted in this background. If H-NS is essential, authors can consider depleting the protein, or combining the H-NS deletion with *rpoS* mutants that provide a partial rescue to the H-NS null phenotype.

In *S. Typhimurium*, *hns* deletion mutants are very difficult to obtain and they are associated with a strong fitness cost and the acquisition of suppressor mutations (Ali *et al.*, 2014). In *S. Typhimurium* SL1344, the strain used in the manuscript, a *hns* mutant is not possible to be constructed without deleting *rpoS* first (Battesti *et al.*, 2012). We decided not to use the double deletion mutant *hns rpoS* because it is broadly documented that *rpoS* mutants are attenuated and this could alter the remodeling of the SPI-1 chromatin landscape due to a dysregulation on SPI expression, leading to a poor SPI-1 transcriptional activation. Additionally, using a H-NS depletion system will require a considerable time of development that will strongly delay the revision of this article. We therefore studied SPI-1 expression dynamics in physiological conditions (WT background). To determine the role of H-NS in SPI-1 chromatin remodeling, we determined the binding occupancy of H-NS in SPI-1 silenced or active populations. Our results show that the key role of H-NS is to inhibit transcription, in agreement with previous studies (Navarre *et al.*, 2006; Lucchini *et al.*, 2006) (Figure 2, Figure 3, Figure 4 and new Supplementary Figure 6).

To further support this,

[redacted]

[redacted]

Altogether, these news results, in conjunction with those already present in the previous version of this manuscript, show that H-NS main role is to repress transcription, while HilD binding plays a key role in SPI-1 activation (or chromatin remodeling through transcription activation). However, since HNS-1 results rise new questions concerning the role of the DNA binding domain on H-NS repression (i.e., which are the interacting partners involved in repressing SPI-1 activity in the presence of HNS-1?), they will be not included in the revised version of the manuscript and will be part of future and detailed analyses. Of note, during the revision of this article a new study in *E. coli* using Micro-C supported these conclusions by showing that the main role of H-NS, in concert with StpA, is to promote bridged structures that are associated with repressed chromatin (Gavrilov *et al.*, 2025).

[redacted]

[redacted]

[redacted]

[redacted]

4. Section on spurious transcription from H-NS silenced SPI chromatin lacks clarity and experimental strength. It is clear in Supplementary Figure 5 that genes occupied by H-NS has lower sense and antisense transcription. But to identify transcriptional changes specific to pathogenicity islands, sense and anti-sense activity of HNS-bound pathogenicity islands should be compared to the rest of HNS-bound genes and HNS-free genes. This will clarify if the spurious antisense transcription observed here is unique to these pathogenicity islands. For this comparison genes from all pathogenicity islands could be pooled to avoid clusters with very low gene numbers. For example, in Figure 3, the comparisons are made between clusters ranging from 3 genes to >4000 genes. In addition, the logic for using highly expressed genes is vague. If the authors maintain this cluster, the logic should be stated in text and how HEGs are defined should be clearly stated.

We thank the reviewer for this comment. We have re-analyzed the sorted RNA-seq comparing SPI genes vs non-SPI genes as proposed by this reviewer. These new analyses show that in repressive conditions, sense transcription is significantly lower in SPIs compared to non-SPI genes, regardless of whether the SPI genes are bound by H-NS or not. Strikingly, antisense transcription is significantly higher in SPI genes that are not bound by H-NS, when compared to non-SPI genes. In summary, **these results suggest that silent SPI present a specific transcriptional landscape**

compared to non-SPI genes, with H-NS playing an important role in decreasing spurious antisense transcription.

Altogether, these new analyses are consistent with the role of H-NS in decreasing spurious transcription, and highlight a high antisense transcription in silenced SPIs genes not bound by H-NS, likely due to their high AT content (See reviewer 3 comment). These results will be part of the revised section starting in L161, and the new figure will replace previous version of Supplementary Figure 5.

New Supplementary Figure 5. Level of transcription in non-SPI and SPI genes in GFP⁻ and GFP⁺ cells in ESP.

- a Antisense transcriptional activity within non-SPIs (turquoise) and SPI (red) genes in GFP⁻ cells in ESP.
- b Sense transcriptional activity within non-SPIs (turquoise) and SPI (red) genes in GFP⁻ cells in ESP.
- c Antisense transcriptional activity within non-SPI genes and SPI genes either free of H-NS (light blue) or bound by H-NS (yellow) in GFP⁻ cells grown in ESP.

- d Sense transcriptional activity within non-SPI genes and SPI genes either free of H-NS (light blue) or bound by H-NS (yellow) in GFP⁻ cells grown in ESP.
- e Antisense transcriptional activity within non-SPIs (turquoise) silent SPIs (red) and active SPIs (pink) in GFP⁺ cells in ESP.
- f Sense transcriptional activity within non-SPIs (turquoise) silent SPIs (red) and active SPIs (pink) in GFP⁺ cells in ESP.
- g In all the panels, pairwise comparisons were performed using the Wilcoxon rank sum test with continuity correction to assess the statistical significance ($p < 0.05$). The number of analyzed genes ('#') per feature is also indicated.

Finally, the logic for using highly expressed genes (HEGs) is that for almost a decade it has been known that their activity is critical in chromosome conformation (Le and Laub, 2016) and they are found associated with TIDs (Bignaud *et al.*, 2024). In addition, their high level of expression will allow the reader to have a qualitative idea of the level of SPI transcriptional activity. In the revised version, the main text will now say (L185 to L188):

“As expected, in active SPIs and in highly expressed genes that are frequently associated with TIDs and not related to SPIs (HEGs), sense transcriptional activity is significantly higher than in RoGs and AS activity is decreased (Figure 3B and Supplementary Figure 5F).”

5. The significance of the SPI-I locus mobility to nucleoid periphery is unclear. How do the authors propose this occurs? Authors should also provide imaging of a control region not within the SPI-1 locus to show that this observation is SPI-1 specific. Additionally, this mobility should increase in hilE deletion and be abrogated in hilD deletion.

We are confused with this question since as stated in the discussion section we propose that (L317 to L323):

“The differences in subcellular localization of active and silent SPI-1 lead us to propose that the relocation of SPI-1 towards the nucleoid periphery is the result of high transcription, likely enhanced by the coupling of transcription with translation (Yang *et al.*, 2019) of the components of the T3SS1 inserted into the cytoplasmic membrane. This interpretation is supported by results in *Vibrio parahaemolyticus* that show a genetic locus encoding a type three secretion system captured at the inner membrane where it is expressed and the T3SS is assembled (Kaval *et al.*, 2023), and by localization of active RNA polymerase at the nucleoid periphery in *E. coli* (Stracy *et al.*, 2015).”

In order to show that this is specific to SPI-1 activation, we have studied the subcellular localization of the unrelated and silent *yggF* locus, in active SPI-1 and silent SPI-1 cells grown in ESP, by inserting the *parS^{PMT1}* site downstream this gene. This locus is placed at 208 kb downstream of SPI-1. Results are shown in the revised version of Figure 5 (**Revised Figure 5**) and in Supplementary Figure 9 (**Revised Supplementary Figure 9**). Now the revised text says (L266-L275):

To confirm that changes in subcellular localization were specific to active SPI-1, we analyzed the position of a silent chromosomal locus unrelated to SPI-1, located 208 kb downstream of this pathogenicity island. Specifically, the *parS^{PMT1}* was inserted after the stop codon of the gene *yggF*, which is silent under the ESP condition (Supplementary Data 1 and Supplementary Figure 9C). The position of *yggF* was analyzed in a strain carrying the transcriptional fusion of the gene *prgH* with the *mCherry* gene at its native position in the SPI-1 locus, to determine the localization of this locus when SPI-1 was silent (*mCherry*⁻) or active (*mCherry*⁺). We did not observe a significant difference in the localization of ParB:YFP/*parS^{PMT1}* in *mCherry*⁺ and *mCherry*⁻ cells (Figure 5D), neither a difference between the percentage of ParB:YFP/*parS^{PMT1}* foci that were detected outside the segmented nucleoids in *mCherry*⁻ or *mCherry*⁺ cells (Supplementary Figure 9D).'

Revised Figure 5. Subcellular localization of SPI-1 in *Salmonella* cells.

- a** Upper panel: schematic representation of the SPI-1 locus, showing the *prgH*(stop)*mCherry* transcriptional fusion in its native location and the insertion of *parS^{PMT1}* site, 1 kb upstream of the last SPI-1 gene. Lower panel: Combination of SIM² images of *Salmonella* nucleoids (Hoescht), ParB:YFP/*parS^{PMT1}* (YFP), wide-field signal of *mCherry*⁺ and a merge of the three images (maximum intensity projection) in ESP. Scale bar = 2 μm.

- b Representative 3D SIM field view before and after segmentation and identification of active SPI-1 cells in ESP. After segmentation, nucleoids of mCherry⁻ cells are shown in cyan, nucleoids of mCherry⁺ in red and the SPI-1 locus as a green dot. Scale bar = 2μm.
- c The distance of ParB:YFP/*parS*^{DMT1} focus is measured from cells by taking the shortest distance to the segmented nucleoid surface, with positive values (+*d*) representing distances inside the segmented nucleoid (high DNA density regions) and negative distances (-*d*) representing distances outside the segmented nucleoid.
- d Box plot representing the distribution of distances for a given ParB:YFP/*parS*^{DMT1} focus to the nucleoid surface (black line, '0' value) in mCherry⁻ (yellow) and mCherry⁺ (pink) cells, at the SPI-1 locus (left panel) or the *yggF* locus (right panel). Positive y-axis values represent distances measured for foci that are in the segmented nucleoid (blue), negative values represent distances measured for foci that are outside the segmented nucleoid (gray).

Concerning the effect of *HilE* and *HilD* on SPI-1 localization, our results show that deleting *hilE* increase the proportion of SPI-1^{ON} cells (from ~10% to ~29%, Table 1). Then we do not expect an increase in the mobility of the SPI-1 locus when activated in this mutant, but a higher number of cells presenting SPI-1 closer to the nucleoid periphery. Furthermore, the 3D folding of the chromatin in *hilE* mutant is similar to that of WT cells (Supplementary Figure 4), therefore, and to our understanding, the analysis of SPI-1 localization in this mutant is not relevant. Regarding *HilD*, in the original version we have provided data that show that in the absence of *hilD*, SPI-1 is repressed (Table 1) and TIDs are not formed (Figure 2). Consequently, in the absence of transcriptional activation it is not expected to observe a relocation towards the nucleoid periphery.

Revised supplementary Figure 9 ParB:YFP/*parS*^{DMT1} foci localization in *Salmonella* cells

- a Computation of the *p*-value, obtained from a Mann-Whitney test applied to the ParB:YFP/*parS*^{DMT1} foci distances from the nucleoid surface in both mCherry⁻ and mCherry⁺. To avoid a side effect of

the population size, we used the same number of foci for each population (n=652). To do so, the distances of all the mCherry⁺ foci were compared to a random subpopulation of mCherry⁻ foci (repeated 10,000 times). This approach demonstrates that disparities in population size do not influence the significance of the difference in localization between the two populations.

- b Histogram representing the fraction of the ParB:YFP/*parS*^{pMT1} foci at SPI-1 locus detected outside the segmented nucleoid region, in both mCherry⁻ and mCherry⁺ populations grown in ESP condition. Chi-square test showing the significance of the differences between the two populations (p-value = 2.485e-16).
- c Histogram representing the fraction of the ParB:YFP/*parS*^{pMT1} foci at *yggF* detected outside the segmented nucleoid region, in both mCherry⁻ and mCherry⁺ populations grown in ESP. Differences are not significant.

6. The model that the authors have proposed is speculative. A particular observation that needs further validation is the role of Spot-42 srRNA, which is brought up recurrently in the manuscript as being responsible for HilD positive regulation. In addition, the upregulation of this Spot-42 gene *spf* contrasts with previous studies, and hence it would be a key experiment to test the importance of this upregulation. Given the model proposed by the authors, expression of Spot-42 in exponential phase would induce SPI-1 bistable expression and absence of the srRNA could lead to lack of such bistable expression in stationary phase. More interestingly what could be the impact of depleting this srRNA on formation of TIDs. Without further experimental evidences, it is recommended that the authors tone down the speculative nature of the model.

To address this reviewer comments, we have deleted *spf*, but we observed minor changes in SPI-1 expression (data not shown). The absence of a major phenotype of this deletion could be linked to the activity of another srRNA acting on *hilD* mRNA (Abdulla *et al.*, 2022). Nonetheless, we do not find our results to be in conflict with those previously published by Hinton's lab since these experiments were carried out in heterogenous populations in which only a fraction of cells expressed SPI-1 (ESP condition). To confirm this, we performed an **in-silico analysis** by subsampling and merging RNA-seq FASTQ files from sorted GFP⁺ and GFP⁻ populations to reconstruct a mixed population (i.e., ~10% GFP⁺ + ~90% GFP⁻) in replicates. We then reanalyzed these merged datasets together with the RNA-seq dataset from Kröger's study (Kröger *et al.*, 2013), and found that *spf* in the reconstructed ESP heterogenous population is repressed (Fold Change Rep1 (log2) = -10.51; Fold Change Rep2 (log2) = -10.36), comparable to Kröger's experimental data (Fold Change Rep1 (log2) R1 = -4.39; Fold Change Rep2 (log2) = -4.56). These results indicate that the upregulation of *spf* seems to be specific to GFP⁺ cells and is 'diluted' in the heterogeneous ESP population.

In summary, we have toned down the speculative nature of the model, removed *spf* from our model in Figure 6 (see Revised Figure 6) and from the discussion section of the revised version of the manuscript, which now says (L331-L332):

'In contrast, in the GFP⁺ population, *hilD* mRNA is stabilized, potentially due to its interaction with Spot-42 associated to Hfq (Mouali *et al.*, 2018).'

Revised Figure 6. Model of SPI-1 chromatin remodelling upon expression.

- a In repressed SPI-1 cells in ESP, SPI-1 chromatin is localized far from the nucleoid periphery, shows a high H-NS occupancy and is permeable to low levels of RNA polymerase molecules that promote the spurious transcription of SPI-1 promoters, such as those of *hilD*. Spurious *hilD* transcription is rapidly degraded. If *hilD* mRNA escapes degradation, the translated HiID protein will form a heterodimer with HiIE, impeding the activation of SPI-1.
- b In a fraction of cells in ESP, translation of *hilD* mRNA triggers a feedback loop that activates SPI-1. The binding of HiID to its own promoter leads to a conformational change that promotes the unloading of H-NS, mainly from coding regions, allowing the correct positioning of RNA polymerase and transcript elongation. This is accompanied by the folding of SPI-1 chromatin into several domains and a relocalization close to the nucleoid periphery, where it will facilitate access of RNA polymerase, transcription, translation, assembly, and quick repression of this T3SS.

Minor comments:

1. Order of figures should match their citation in the text.

Checked. Thank you for highlighting this.

2. Please label the conditions for each figure panel in the figures. Presently, it is very confusing to know which panel corresponds to which experimental condition.

We have now included the legend 'GFP⁺ cells in ESP' or 'GFP⁻ cells in ESP', where required. In addition, we have clearly stated this in the figure legends in the revised version of the article.

3. Please provide information on the number of replicates for each experiment.

For genomic experiments, this information was present in Supplementary Tables 6, 7 and 8 (Supplementary Tables 5, 6 and 7 in the revised version of the manuscript) and in the GEO browser under the accession number GSE289504. We have also clarified this in L498-L500:

'All libraries generated in this study, including the number of replicates, are listed in Supplementary Table 5, Supplementary Table 6 and Supplementary Table 7.'

And in L592-L594:

'For SPI-1 localization, two independent experiments were combined and analyzed together, whereas a single replicate was analyzed for *yggF* localization.'

4. It would be interesting to see if there is a correlation between mCherry intensity (indicating the extent of SPI-1 activation) and the distance of the genetic locus?

Unfortunately, it is not possible to make a link between the intensity of mCherry and the subcellular localization of SPI-1 expression because mCherry intensity is a limiting factor for quantification in the studied conditions. The ideal condition for optimal mCherry fluorescence is minimal medium and 30°C, and cells are grown in LB at 37°C. Furthermore, half-life of mCherry may also bias the interpretation of the results. For these reasons, we prefer not to perform this analysis.

5. L45. "de-repressed" instead of "depressed"

The reviewer may refer to line 46 'derepression'. The word is well spelled and we will not change it.

6. L98: You should elaborate on the 39 genes - how many of them are SPI-1 and non-SPI-1 and what proportion of SPI-1 genes have been identified as upregulated.

This information was present in Supplementary Table 2 and 3 (new supplementary Table 1 and 2). However, following this reviewer suggestion, we have clarified this information in the revised version (L98-101):

'Furthermore, 39 of the upregulated genes in GFP⁺ cells correspond to SPI-1 genes (30 genes) or other SPI genes (SPI-5: 2 genes; SPI-11: 2 genes, SPI-2: 1 gene; SPI-3: 1 gene; SPI-4: 4 genes), most of them known to be co-regulated with SPI-1 (Kröger *et al.*, 2013) (Figure 1, Supplementary Table 1 and Supplementary Table 2, Supplementary Data 1).

7. L128: Change 'observed' to 'observe'

Changed

8. L144: The significance of the observation about TIDs encompassing *hilA* and *invF* needs to be stated here. For example, no change in H-NS occupancy is observed. Could you comment on this?

Line 144 (now L147) refers to ChIP-seq results obtained using a bistable population of *Salmonella* (Supplementary Figure 3) and as mentioned in lines 116 to 119, TIDs are observed only when Hi-C is performed on sorted GFP⁺ cells in ESP. Therefore, it is logical that we observed no changes in H-NS occupancy in non-sorted and bistable SPI-1^{ON/OFF} populations (Supplementary Figure 3). Furthermore, it is difficult to interpret the significance of these sets of ChIP-seq data since the conditions are not the same. We just described our results without further assumptions. We addressed this point when discussing the ChIP-seq of H-NS in the same condition in which TIDs were observed (see section starting in L193). To avoid any confusion, we have clarified this in the revised version of the article (L144-L147):

‘When comparing the binding occupancy of the master activator HiiD in a non-sorted, bistable SPI-1^{ON/OFF} population (ESP condition, Supplementary Figure 3), we observed that the genetic region encompassing the SPI-1 *hilA* and *invF* genes, correspond to the two regions with higher HiiD occupancy.’

9. L205-210. RNA polymerase itself could be a source of H-NS dissociation from these loci.

We thank the reviewer for this comment. The revised version now says (L215-L218):

‘...These results suggest that SPI-1 chromatin might be bound by RNA polymerase and/or specific regulatory factors that compete with H-NS binding, leading to a significant reduction in its binding and allowing sustained expression of SPI-1 genes and radical chromatin remodeling.’

10. Fig. 4C-D. It is unclear what the reader is supposed to focus on. Could you please indicate via arrows/ lines?

We have clarified this by stating that the reader should focus on the MACS2 result below the ChIP-seq. The new text is the following (L220-L221):

‘MACS2(Zhang *et al.*, 2008), a tool developed for identifying DNA regions bound by DNA binding proteins, revealed that H-NS binding in this population shows a variable pattern of occupancy along the SPI-1 locus (Figure 4C-D)’

11. L220: There is no Figure 4F as cited in text.

Changed to 4E

12. L234: It is unclear what you mean by ‘assure a source of repressor’

This reviewer is right, this was unclear and it was deleted.

13. L257-259 is speculative and should be removed unless data to support this conclusion is provided.

We respectfully disagree with the reviewer’s interpretation. We have shown that when SPI-1 is expressed, TIDs are formed (Figure 2) and that when SPI-1 is expressed, this locus is localized closer to the nucleoid periphery (Figure 5). We have included new data that shows that this re-localization is specific of active SPI-1 (Revised Figure 5, see above). Therefore, our sentence is not speculative, it is just describing two results of our article.

14. L365: The plasmid pGBD3.2 is not listed in the plasmid sheet

We thank this reviewer for highlighting this mistake. The plasmid used is known as pGBKD3-parSpmT1 and now it is correctly cited in Supplementary Table 5.

15. Construction of pMKB1 and pSPB5 should be included.

We included details of the construction of plasmid pMKB1 in L385-L389 and pSPB5 in L392-L393. We also modified Supplementary Table 5 accordingly.

16. L489: Spelling of cell pellets needs correction

We corrected this spelling error.

17. Fig. 3G legend: Spelling of beginning needs correction

We have corrected this error.

18. Fig. 4A: Show a representative 1 kb region with H-NS occupancy and contact signal outside SPI-1

We are not sure that we understand this comment. Figure 4A shows a ~1 Mb zoom-in of a whole-genome correlation between Hi-C contacts and H-NS occupancy, both calculated at 1-kb bins, including non-SPI-1 regions.

19. Supplementary Figure 5 legends - Clarify what Category 1 is.

We have removed this figure from the revised version of the manuscript due to comment #4. However, we have better described how we defined HEGs, since this is related to the classification of gene expression into different category levels. The revised manuscript now says (L650-L657)

‘To classify gene expression levels, the normalized counts generated using the SARTools DESeq2 pipeline were further adjusted for gene length (calculated as DESeq2 normalized reads divided by gene size and per million reads, transcripts per million (TPM)). This allowed us to compare directly the relative levels of gene expression of individual genes. Expression levels were classified from poorly expressed to highly expressed as previously done in (Lioy *et al.*, 2021). Specifically, poorly expressed genes are defined as those whose expression levels fall within the 25th percentile of the expression distribution across the entire chromosome. Highly expressed genes (HEGs) are defined as those whose expression levels lie between the third quartile (Q3) and the maximum observed expression value across the chromosome.’

Reviewer #2 (Remarks to the Author):

Reviewer #3 (Remarks to the Author):

In this paper Kordei et al use a combination of global and microscopic methods to study H-NS and transcriptional silencing in Salmonella. The experiments are done to a high standard, and I have no issue with anything in results sections 1, 2 or 5. However, I found the way the story is presented in results sections 3 and 4 quite confusing and I wonder if some of the data can really be interpreted as described. My comments are quite lengthy because I'm trying to address some important but subtle points. I hope the review is clear and useful to the authors.

SECTION BEGINNING LINE 159: I found the way this section was presented to be very confusing. I think this is largely because of the way the rationale for the experiments is set up. The paragraph starts by suggesting a discrepancy between existing stories in the literature (i.e. H-NS repressing

pervasive transcription vs some pervasive transcription being evident at H-NS bound loci). My personal interpretation is that the two stories are consistent with each other; H-NS deletion permits high-level spurious transcription, but some is still detectable, albeit at much lower levels, when H-NS is present. My recollection is that both papers, and several more recent studies, show data consistent with this. There's quite a lot to pull apart here so I've tried to separate the issues below.

1. Assuming there is some sort of disagreement, which I don't think there is, the authors then describe experiments that don't really discriminate between the two possibilities. Instead, they compare levels of spurious transcription within H-NS bound regions, and other parts of the chromosome, in GFP- cells. The conclusion seems to be that H-NS represses spurious transcription but that these levels of spurious transcription are higher than expected for some H-NS bound SPIs*. I'm not suggesting the authors do it but, if they really think there is some sort of disagreement in the literature, surely the key experiments would be to: i) test H-NS +/- cells to determine effects on spurious transcription or ii) do the experiment in GFP+ cells to determine if levels of spurious transcription increases when the SPIs are turned on (a caveat is that I don't think RNA-seq easily measures spurious transcription, see below).

(*to my mind the key question is whether this transcription is truly occurring when H-NS is still bound, implying some sort of chromatin remodeling, or whether H-NS is being stochastically released, and/or the SPIs turned on, within the population of GFP- cells. I also wonder if these SPIs are unusually AT-rich, compared to other H-NS bound loci, which might account for the higher spurious transcription)

We thank to this reviewer the detailed comment. We agree that the introduction to this section was confusing and we have changed it to (L163-L167):

“It has been proposed that the main function of H-NS is to suppress intragenic transcription events (Singh *et al.*, 2014; Lippa *et al.*, 2021). Furthermore, a recent study in *Salmonella* showed that, even under repressed conditions, SPI-1 presents a significant level of pervasive transcription originating from antisense promoters. This leads to an open window for displacement of H-NS from regulatory regions, allowing HilD to bind its own promoter and activate SPI-1 transcription (Figueroa-Bossi *et al.*, 2022) “

In addition, we have performed new analyses shown in the new Supplementary Figure 5, that describe the role of H-NS in silencing spurious antisense transcription.

2. Lines 174-178: “Altogether, these results confirm that high H-NS occupancy co-exists with spurious sense and antisense transcriptional activity at repressed SPI-1 and other silent SPIs (Figure 3, blue boxplots). This is further supported by the low RNA polymerase occupancy within repressed SPIs observed in non-sorted exponentially growing *Salmonella* cells (Figure Supplementary 6).” I don't follow the authors logic here, why does the low RNAP occupancy support the presence of spurious transcription?

We thank the reviewer for highlighting this, we removed this sentence and the figure from the revised version.

3. I think there may be differences between definitions of transcriptional events used here and in prior papers. There are also differences in exactly what is measured by different experimental tools in the various papers. In this work, the terms “pervasive” and “spurious” transcription are used interchangeably, unless I misunderstand. Obviously, there’s no formal definition, but to me the meanings are subtly different. I would describe pervasive transcription as a “catch all” term referring to the observation that RNAs are made everywhere, and on both strands, to some extent. There are two main causes of this (transcription initiation in unusual locations and inefficient termination of mRNAs). Spurious transcription, at least as I see it, refers to the sub-category of initiation in unusually locations and, more specifically, the propensity of RNAP to initiate from many sites in AT-rich DNA. With respect to exactly what is measured, RNA-seq will find reasonably abundant RNAs and detect some RNAs resulting from spurious transcription. However, an issue is that spurious sense transcripts, from within genes, can be hidden by the overlapping signal from the full-length mRNA. Hence, TSS mapping also allows for much easier differentiation between mRNAs and spurious intragenic sense RNAs. A further issue is that spurious RNA’s get terminated quickly by Rho. Again, this means spurious RNAs are easier to find if only the TSS is mapped, instead sequencing total RNA.

We thank this reviewer for this kind explanation, we agree that it is difficult to define ‘pervasive’ and ‘spurious’ without further experiments. However, we agree that ‘spurious’ will be a more accurate term. We have now replaced ‘pervasive’ by ‘spurious’ and we have included a sentence to define ‘spurious transcription’ in our work (L167-L169):

‘To further explore the transcriptomic data obtained from GFP⁻ population, we decided to investigated the level of spurious transcription—defined as the level transcription in sense or antisense direction that arise in repressed conditions— in SPIs genes (Figure 3)’

4. To summarize the above, I think all existing data are consistent with H-NS repressing spurious transcription, but not to the point where it becomes undetectable. If the authors agree that this is what the existing data tell us, it sounds like the interesting result here is that some SPIs have unusually high levels of background spurious transcription, when H-NS is present. I think the section would make a lot more sense if presented in this way.

We agree with all the comments raised by this reviewer and we are sorry if this section was not clear. To avoid any confusion, we have removed these sentences, we have refined our analyses and included a new **Supplementary Figure 5** that address the comments raised by this reviewer, in which we show that SPIs present a high level of spurious antisense transcription in H-NS free genes (Please, see answer to comment #4 of R1).

SECTION BEGINNING ON LINE 184: I think this is the key question the authors skirt around in the section discussed above; does transcription (spurious or not) evict H-NS from the DNA? I think this is a very difficult question to answer unequivocally and would require a single cell approach. Presumably, the populations of GFP⁻ and GFP⁺ cells collected are not truly homogeneous (i.e. GFP⁻ and + interconvert, at low levels, all of the time).

We agree with these comments and we show in the Supplementary Figure 1 of original version of the manuscript that after sorting the GFP⁺ population is not homogeneous. In addition, this was already mentioned in the main text (L90-L91)

1. The authors show that a silent SPI-1 exhibits low level spurious transcription and high-level H-NS binding. I think there are several plausible explanations for this but all fall into one of two broad models: i) RNAP can make spurious RNAs at low levels when H-NS is bound, perhaps implying local chromatin rearrangements or ii) in a small number of GFP⁻ cells, H-NS is transiently released from the DNA and transcription occurs. At the population level, both models would be consistent with high H-NS binding but low transcription. I'm not sure the approach here can differentiate between the two possibilities.

We agree with this reviewer that we cannot differentiate between the two possibilities, and we have refined our analyses since our objective was not to address this point since the GFP⁺ population is not homogeneous. However, in the GFP⁻ population, we do have a homogeneous population and we revealed that spurious antisense transcription is associated with H-NS free genes in SPIs. This increased antisense transcription seems to be unique of SPIs (**New Supplementary Figure 5**)

2. The authors show that full induction of SPI-1 (in GFP⁺ cells) leads to reduced (but not abolished, see point 3 below) H-NS binding. The implication is that transcription does evict H-NS, which I can believe. However, it's difficult to know what gives rise to the residual H-NS binding signals. Does H-NS truly remain bound? Does this signal come from a low background of cells that have reverted to the GFP⁻ phenotype? Does H-NS transiently release the DNA and rebind in such a way that can't be detected at the level of the whole GFP⁺ population?

We agree with this reviewer and in order to answer these questions, we would need single cell approaches. But we can speculate that the reduced levels of H-NS, at least in the regions in which HilD is bound (Supplementary Figure 3) may come from the low level GFP⁻ population.

3. I don't agree with the statement that "H-NS-free regions were found all along the central region of SPI-1, the largest ones at the hilA and invF regulon, specifically within the genes orgB, orgA, prgK, prgJ, part of prgH, invG". I would say that H-NS binding is reduced but remains higher than background levels since elsewhere in Figure 4D.

We kindly disagree with this remark, because H-NS free regions were detected using MACS2 (Zhang *et al.*, 2008). This algorithm is the standard in the field to detect regions of the chromosome that are statistically enriched by a DNA binding protein compared to the control condition (INPUT sample in this study). In figure 4D, these peaks are highlighted as yellow bars below the H-NS ChIP-seq. The bar-free regions are therefore, H-NS free regions. The corresponding peaks revealed by MACS2 were present in Supplementary Data 2.

OVERALL THOUGHTS

This is a very nice piece of work; the results in Figures 1 and 2 are great and reveal expected results. i.e. increased transcription generates TIDs and H-NS bound regions have very low levels of long-distance contacts (indicating bridging). Similarly, movement of transcribed SPI-1 to the periphery of the nucleoid is nicely demonstrated in Figure 5, again as expected. With respect to what is truly new/unexpected, the major advance would be to determine if H-NS is fully evicted from transcribed regions or remains associated in some way. In this respect, I don't think the data presented can provide a complete answer. Whilst the model in figure 6 is very plausible, there are other interpretations. It's also notable that the model isn't particularly different to that shown in figure 6 of Figueroa-Bossi et al. That said, the authors do improve on the Figueroa-Bossi paper by better separating out the H-NS binding pattern in SPI-1 induced vs uninduced cells.

We thank to the reviewer for the positive comment. However, while we appreciate the reviewer's opinions, we would like to clarify some points:

1. H-NS binding regions do not show low levels of long-range distance contacts, indeed H-NS bound regions behave as any other H-NS free and silent region in which long-range contacts are not affected. Therefore, our results do not indicate bridging. It is important to mention, that during this article was in revision, a new technique based in micrococcal MNase (Micro-C), showed clearly bridged structures associated to H-NS in *E. coli*. It could be possible that these structures exist in *Salmonella*, but they would be only revealed using Micro-C (Gavrilov *et al.*, 2025).
2. Active SPI-1 shows low levels of long-range DNA contacts due to the formation of TIDs, which are regions characterized by increased short-range interactions. As previously mentioned and shown in the article, active SPI-1 is associated with (i) a new protein landscape, (ii) a distinct 3D folding, and (iii) a new subcellular localization. These results might have been expected; however, to our knowledge, this is the first time that such findings have been clearly demonstrated under physiological conditions, and specifically, the first time they have been shown for SPI-1.
3. As explained previously, although our SPI-1 ON populations are not homogeneous, our results show for the first time that H-NS is evicted from several SPI-1 genes, as revealed by ChIP-seq and MACS2 analyses. We cannot draw conclusions about the specific regions where H-NS occupancy decreases, as this could be related to the small proportion of GFP⁻ cells within the GFP⁺ population. However, it is possible that the H-NS-free regions are in fact larger than observed.
4. We agree with the reviewer that our results provide a pathway to a new model for active SPI-1.

Minor comments:

Lines 43-46: Is it worth mentioning temperature/osmolarity?

We included this in the introduction of the revised version (L44):

'Reversing the transcriptional silencing activity of H-NS can be achieved by physico-chemical factors (i.e., osmolarity, pH or temperature)(Rashid and Dame, 2024), antagonistic proteins that interfere with the H-NS/DNA filaments, or through alteration of DNA topology (bending or supercoiling), leading to specific derepression(Stoebel *et al.*, 2008; Grainger, 2016; Hustmyer and Landick, 2024; Rashid and Dame, 2024; Figueroa-Bossi *et al.*, 2024).'

PMID: 33245158 is probably worth a mention regarding H-NS silencing of intragenic promoters.

Mentioned in line 163.

Title and various places in the paper: I think it would be helpful to define what is meant by remodeling. To me, this means that the chromatin remains largely intact but is altered. This is a personal opinion, but if H-NS is completely evicted by transcription I would not define this as remodeling.

We included a definition for chromatin remodeling in L153-L155:

‘In the absence of HilD (Golubeva *et al.*, 2012), SPI-1 is not active and TIDs are not formed (Table 1, Figure 2F), confirming that HilD is necessary for SPI-1 chromatin remodeling (i.e., changes in the protein landscape and local 3D organization) and the formation of TIDs.’

Line 280: “We did not observe specific bridging between or within repressed SPI loci”. Presumably it’s impossible to say, at this resolution, if you really see short range bridging (i.e. between H-NS molecules bound to DNA sites separated by less than ~2 kb). Worth mentioning?

We looked for specific bridging between silenced SPIs at different resolutions (1kb, 2kb and 5kb) and we did not observe any. However, as mentioned above, and discussed in the revised version of this article (L301-L306), it could be likely that Hi-C cannot capture these bridged structures and Micro-C (Gavrilov *et al.*, 2025) is the pertinent technique to study H-NS silent chromatin.

References.

- Abdulla, S.Z., Kim, K., Azam, M.S., Golubeva, Y.A., Cakar, F., Slauch, J.M., and Vanderpool, C.K. (2022) Small RNAs Activate Salmonella Pathogenicity Island 1 by Modulating mRNA Stability through the hilD mRNA 3' Untranslated Region. *J Bacteriol* e0033322.
- Ali, S.S., Soo, J., Rao, C., Leung, A.S., Ngai, D.H.-M., Ensminger, A.W., and Navarre, W.W. (2014) Silencing by H-NS Potentiated the Evolution of Salmonella. *PLoS Pathog* **10**: e1004500.
- Battesti, A., Tsegaye, Y.M., Packer, D.G., Majdalani, N., and Gottesman, S. (2012) H-NS Regulation of IraD and IraM Antiadaptors for Control of RpoS Degradation. *J Bacteriol* **194**: 2470–2478.
- Bignaud, A., Cockram, C., Borde, C., Groseille, J., Allemand, E., Thierry, A., *et al.* (2024) Transcription-induced domains form the elementary constraining building blocks of bacterial chromosomes. *Nat Struct Mol Biol* .
- Figueroa-Bossi, N., Fernández-Fernández, R., Kerboriou, P., Bouloc, P., Casadesús, J., Sánchez-Romero, M.A., and Bossi, L. (2024) Transcription-driven DNA supercoiling counteracts H-NS-mediated gene silencing in bacterial chromatin. *Nat Commun* **15**: 2787.
- Figueroa-Bossi, N., Sánchez-Romero, M.A., Kerboriou, P., Naquin, D., Mendes, C., Bouloc, P., *et al.* (2022) Pervasive transcription enhances the accessibility of H-NS-silenced promoters and generates bistability in Salmonella virulence gene expression. *Proc Natl Acad Sci U S A* **119**: e2203011119.
- Gavrilov, A.A., Shamovsky, I., Zhegalova, I., Proshkin, S., Shamovsky, Y., Evko, G., *et al.* (2025) Elementary 3D organization of active and silenced E. coli genome. *Nature* 1–11.

Golubeva, Y.A., Sadik, A.Y., Ellermeier, J.R., and Slauch, J.M. (2012) Integrating Global Regulatory Input Into the Salmonella Pathogenicity Island 1 Type III Secretion System. *Genetics* **190**: 79–90.

Grainger, D.C. (2016) Structure and function of bacterial H-NS protein. *Biochem Soc Trans* **44**: 1561–1569.

Grenz, J.R., Cott Chubiz, J.E., Thaprawat, P., and Slauch, J.M. (2018) HilE Regulates HilD by Blocking DNA Binding in Salmonella enterica Serovar Typhimurium. *J Bacteriol* **200**: 10.1128/jb.00750-17.

Hautefort, I., Proença, M.J., and Hinton, J.C.D. (2003) Single-Copy Green Fluorescent Protein Gene Fusions Allow Accurate Measurement of Salmonella Gene Expression In Vitro and during Infection of Mammalian Cells. *Appl Environ Microbiol* **69**: 7480–7491.

Hinton, J.C.D., Santos, D.S., Seirafi, A., Hulton, C.S.J., Pavitt, G.D., and Higgins, C.F. (1992) Expression and mutational analysis of the nucleoid-associated protein H-NS of Salmonella typhimurium. *Mol Microbiol* **6**: 2327–2337.

Hustmyer, C.M., and Landick, R. (2024) Bacterial chromatin proteins, transcription, and DNA topology: Inseparable partners in the control of gene expression. *Mol Microbiol* **122**: 81–112.

Kaval, K.G., Chimalapati, S., Siegel, S.D., Garcia, N., Jaishankar, J., Dalia, A.B., and Orth, K. (2023) Membrane-localized expression, production and assembly of Vibrio parahaemolyticus T3SS2 provides evidence for transertion. *Nat Commun* **14**: 1178.

Kröger, C., Colgan, A., Srikumar, S., Händler, K., Sivasankaran, S.K., Hammarlöf, D.L., et al. (2013) An infection-relevant transcriptomic compendium for Salmonella enterica Serovar Typhimurium. *Cell Host Microbe* **14**: 683–695.

Le, T.B., and Laub, M.T. (2016) Transcription rate and transcript length drive formation of chromosomal interaction domain boundaries. *EMBO J* **35**: 1582–1595.

Le, T.B.K., Imakaev, M.V., Mirny, L.A., and Laub, M.T. (2013) High-resolution mapping of the spatial organization of a bacterial chromosome. *Science* **342**: 731–734.

Lioy, V.S., Lorenzi, J.-N., Najah, S., Poinson, T., Leh, H., Saulnier, C., et al. (2021) Dynamics of the compartmentalized Streptomyces chromosome during metabolic differentiation. *Nat Commun* **12**: 5221.

Lippa, A.M., Gebhardt, M.J., and Dove, S.L. (2021) H-NS-like proteins in Pseudomonas aeruginosa coordinately silence intragenic transcription. *Mol Microbiol* **115**: 1138–1151.

Lucchini, S., Rowley, G., Goldberg, M.D., Hurd, D., Harrison, M., and Hinton, J.C.D. (2006) H-NS Mediates the Silencing of Laterally Acquired Genes in Bacteria. *PLoS Pathog* **2**: e81.

Mouali, Y.E., Gavia-Cantin, T., Sánchez-Romero, M.A., Gibert, M., Westermann, A.J., Vogel, J., and Balsalobre, C. (2018) CRP-cAMP mediates silencing of Salmonella virulence at the post-transcriptional level. *PLoS Genet* **14**: e1007401.

Navarre, W.W., Porwollik, S., Wang, Y., McClelland, M., Rosen, H., Libby, S.J., and Fang, F.C. (2006) Selective Silencing of Foreign DNA with Low GC Content by the H-NS Protein in Salmonella. *Science* **313**: 236–238.

Ponndara, S., Kortebi, M., Boccard, F., Bury-Moné, S., and Lioy, V.S. (2024) Principles of bacterial genome organization, a conformational point of view. *Mol Microbiol* .

Rashid, F.-Z.M., and Dame, R.T. (2024) 2024: A “nucleoid space” odyssey featuring H-NS. *BioEssays* **46**: 2400098.

Saini, S., Ellermeier, J.R., Slauch, J.M., and Rao, C.V. (2010) The Role of Coupled Positive Feedback in the Expression of the SPI1 Type Three Secretion System in Salmonella. *PLoS Pathog* **6**: e1001025.

Singh, S.S., Singh, N., Bonocora, R.P., Fitzgerald, D.M., Wade, J.T., and Grainger, D.C. (2014) Widespread suppression of intragenic transcription initiation by H-NS. *Genes Dev* **28**: 214–219.

Stoebel, D.M., Free, A., and Dorman, C.J. (2008) Anti-silencing: overcoming H-NS-mediated repression of transcription in Gram-negative enteric bacteria. *Microbiology* **154**: 2533–2545.

Stracy, M., Lesterlin, C., Garza de Leon, F., Uphoff, S., Zawadzki, P., and Kapanidis, A.N. (2015) Live-cell superresolution microscopy reveals the organization of RNA polymerase in the bacterial nucleoid. *Proc Natl Acad Sci* **112**: E4390–E4399.

Yang, S., Kim, S., Kim, D.-K., Jeon An, H., Bae Son, J., Hedén Gynnå, A., and Ki Lee, N. (2019) Transcription and translation contribute to gene locus relocation to the nucleoid periphery in *E. coli*. *Nat Commun* **10**: 5131.

Zhang, Y., Liu, T., Meyer, C.A., Eeckhoute, J., Johnson, D.S., Bernstein, B.E., *et al.* (2008) Model-based Analysis of ChIP-Seq (MACS). *Genome Biol* **9**: R137.

Ref: *Nature Communications* manuscript NCOMMS-25-20136-T

Manuscript: Bacterial chromatin remodeling associated with transcription-induced domains at pathogenicity islands

Authors: Kortebe et al

Revision date: 2nd December 2025

Corresponding author: Virginia S. Liroy

Point-to-Point response to the reviewers.

We would like to thank the reviewers again for their constructive feedback and for helping to strengthen the manuscript. We have performed the additional changes requested by Reviewer 1 and have addressed the editorial comments. You can find below the point-by-point response to the reviewers comments.

REVIEWERS' COMMENTS

Reviewer #1 (Remarks to the Author):

Overall, authors have presented a thorough revision that addresses most concerns previously raised. I congratulate the authors on this excellent work.

We thank the reviewer for this positive comment

Only a couple of minor typographical errors are highlighted below:

1. L99 - There are only 3 SPI-4 genes as per Supplementary Table 2.

We have corrected line L99. This was an unintentional error.

2. In Supplementary Figure 5, there is no g panel, while it is indicated in legends.

Thank you for highlighting this. We have corrected this typo.

Reviewer #2 (Remarks to the Author):

Reviewer #3 (Remarks to the Author):

The authors have reworked earlier parts of the results section to make the text clearer and better aligned with the existing literature. I think the changes improve the text and make the story clearer. The changes include better defining spurious vs pervasive transcription and a better explanation of exiting stories describing interplay between H-NS and RNAP in coding sequences.

I'm not personally convinced by the description of some SPI regions as H-NS free, but, as the authors describe how they arrive at this description, and readers can easily make up their own minds, I don't feel particularly strongly about this. Whether the authors observe no or low levels H-NS binding doesn't really impact the overall story. Happy to agree to disagree.

With respect to my prior comment that “H-NS bound regions have very low levels of long-distance contacts (indicating bridging)” apologies, this was worded this clumsily. I was trying to say that H-NS bound regions don’t show high levels of long-range contacts and that long range contacts indicate bridging. I agree with the authors’ interpretation of the data.

Overall, the paper is improved and, as on the first occasion, I enjoyed reading it. I’m still left with the feeling that the results are not unexpected, and that there is only a modest advance on what we already know, but that is not a comment on the quality of the work or manuscript. The data support the conclusions and the text puts the work in context of the existing literature.

We thank this reviewer for the thoughtful reassessment of the manuscript and for the clarification regarding long-range contacts and bridging. We are pleased that the revisions improved clarity and alignment with the literature.